



# Analysis of simultaneous aerosol and ocean glint retrieval using multi-angle observations

Kirk Knobelspiesse[1], Amir Ibrahim[1,2], Bryan Franz[1], Sean Bailey[1], Robert Levy[1], Ziauddin Ahmad[1,3], Joel Gales[1,3], Meng Gao[1,2], Michael Garay[4], Samuel Anderson[1,2], and Olga Kalashnikova[4]

[1]NASA Goddard Space Flight Center, Greenbelt, MD, USA
[2]Science Systems and Applications, Inc., Lanham, MD, USA
[3]Science Applications International Corp., Greenbelt, MD, USA
[4]JPL, California Institute of Technology, Pasadena, USA

**Correspondence:** Kirk Knobelspiesse (Kirk.Knobelspiesse@nasa.gov)

**Abstract.**

Since early 2000, NASA's Multi-angle Imaging SpectroRadiometer (MISR) instrument has been performing remote sensing retrievals of aerosol optical properties from the polar orbiting Terra spacecraft. A noteworthy aspect of MISR observations over the ocean is that, for much of the Earth, some of the multi-angle views have contributions from solar reflection by the ocean surface (glint, or glitter), while others do not. Aerosol retrieval algorithms often discard these glint influenced observations because they can overwhelm the signal and are difficult to predict without knowledge of the (wind speed driven) ocean surface roughness. However, theoretical studies have shown that multi-angle observations of a location at geometries with and without reflected sun glint can be a rich source of information, sufficient to support simultaneous retrieval of both the aerosol state and the wind speed at the ocean surface. We are in the early stages of creating such an algorithm. In this manuscript, we describe our assessment of the appropriate level of parameterization for simultaneous aerosol and ocean surface property retrievals using sun glint. For this purpose, we use Generalized Nonlinear Retrieval Analysis (GENRA), an information content assessment (ICA) technique employing Bayesian inference, and simulations from the Ahmad-Fraser iterative radiative transfer code.

We find that four parameters are suitable: aerosol optical depth ($\tau$), particle size distribution (expressed as the fine mode fraction $f$ of small particles in a bimodal size distribution), surface wind speed ($w$) and relative humidity ($r$, to define the aerosol water content and complex refractive index). None of these parameters define ocean optical properties, as we found that the aerosol state could be retrieved with the nine MISR near-infrared views alone, where the ocean body is black in the open ocean. We also found that retrieval capability varies with observation geometry, and that as $\tau$ increases so does the ability to determine aerosol intensive optical properties ($r$ and $f$, while it decreases for $w$). Increases in $w$ decrease the ability to determine the true value of that parameter, but have minimal impact on retrieval of aerosol properties. We explored the benefit of excluding the two most extreme MISR view angles for which radiative transfer with the plane parallel approximation is less certain, but found no advantage in doing so. Finally, the impact of treating wind speed as a scalar parameter, rather than as a two parameter directional wind, was tested. While the simpler scalar model does contribute to overall aerosol uncertainty, it is not sufficiently large to justify the addition of another dimension to parameter space.



An algorithm designed upon these principles is in development. It will be used to perform an atmospheric correction with MISR for coincident ocean color (OC) observations by the Moderate Resolution Imaging Spectroradiometer (MODIS) instrument, also on the NASA Terra spacecraft. Unlike MISR, MODIS is a single view angle instrument, but it has a more complete set of spectral channels ideal for determination of ocean optical properties. The atmospheric correction of MODIS OC data can therefore benefit from MISR aerosol retrievals. Furthermore, higher spatial resolution data from coincident MISR observations may also improve glint screening.

## 1 Introduction

Atmospheric aerosols are one of the largest sources of uncertainty in efforts to understand and predict climate, while the ocean biological and ecological state is a key indicator of ongoing change (IPCC (2013)). Both are systematically observed by spacecraft such as NASA's Earth Observing System (EOS) flagship, Terra. Launched in late 1999, Terra has five instruments with diverse characteristics, including the Multi-angle Imaging SpectroRadiometer (MISR, Diner et al. (1998), Kahn et al. (2005)) and the Moderate Resolution Imaging Spectroradiometer (MODIS, Esaias et al. (1998), Tanré et al. (1997)). These instruments are capable, to varying degrees, of observing the atmospheric and oceanic state using their channels at visible (VIS) and near-infrared (NIR) wavelengths. For example, these observations can be used to identify and characterize suspended particles in the atmosphere (aerosols, e.g. Remer et al. (2005); Sayer et al. (2012); Witek et al. (2019); Garay et al. (2020)), hydrosols in the ocean (representing suspended sediment or phytoplankton, e.g. Mobley et al. (2016), Werdell et al. (2013)), or trace gas absorption in the atmosphere and pigment absorption in the ocean. In many cases, retrieval of aerosol and ocean geophysical parameters for observations over the ocean requires algorithms that account for both, either by simultaneously determining those parameters, approximating their contribution to the observations, or selectively using observations that are minimally impacted by one or the other.

Successful remote sensing requires identification of the geophysical constituents that potentially influence observations, and parameterization of these effects so that they can be incorporated into a retrieval algorithm. The parameterization scheme must be sufficiently representative of the geophysical state and meaningful to the end user, and yet simple enough that it captures the information content contained in an observation and can be practically applied. The retrieval algorithms cited above were derived from generations of experience by the remote sensing community, and are largely based upon avoidance of observation geometries for which sun glint is a potential contributor. However, theoretical studies suggest that incorporation of those observations, along with proper radiometric treatment of glint, can yield more geophysical information. Kaufman et al. (2002) considered a theoretical instrument that makes measurements at two angles, one with sun glint, the other without. Their calculations indicated sensitivity to aerosol absorption, a highly relevant climate parameter. Later, Ottaviani et al. (2013) performed a theoretical study for a multi-angle radiometer with similar (but not identical) characteristics to MISR. They found





not just sensitivity to aerosol absorption, but to other aerosol optical properties, such as $\tau$, real refractive index, and size
distribution. They also found that such a measurement system could simultaneously determine the ocean surface wind speed,
which drives surface roughness and therefore the angular distribution of sun glint (as is modeled in the venerable publication of
Cox and Munk (1954)). Meanwhile, Fox et al. (2007) created an algorithm that retrieves ocean surface wind speed from MISR
observations by minimizing the impact of the atmosphere on measurements. Harmel and Chami, 2012, successfully retrieved
sea surface wind using the polarimetric, multiangle PARASOL (Polarization and Anisotropy of Reflectances for Atmospheric
Sciences Coupled with Observations from a Lidar) instrument, and recently Neukermans et al., 2018 reviewed the use of multi-
angle data for the determination of wind speed. These works indicate that MISR observations can be used to simultaneously
determine aerosol properties and the ocean surface wind speed. Our goal for this paper is to lay the groundwork for such an
algorithm by determining the appropriate level of parameterization for a simultaneous retrieval of aerosol and ocean surface
properties from MISR.

This paper presents an analysis that has two primary tools. First, we use radiative transfer simulations from an iterative
code by Ahmad and Fraser (1982) with the aerosol models described in Ahmad et al. (2010). This code is also the basis of
lookup tables (LUT's) incorporated in the current operational atmospheric correction by the NASA Ocean Biology Processing
Group (OBPG) for ocean color data processing for Sea-viewing Wide Field-of-view Sensor (SeaWiFS), MODIS, Visible In-
frared Imaging Radiometer Suite (VIIRS) and other instruments (Gordon and Wang (1994)). The Ahmad and Fraser Radiative
Transfer (AFRT) code is capable of simulating the aerosol and ocean surface properties that we intend to investigate. Using a
LUT generated by AFRT, we apply the Generalized Nonlinear Retrieval Analysis (GENRA, Vukicevic et al. (2010)) Bayesian
information content assessment technique. Given knowledge of MISR measurement, modeling, and other uncertainties (e.g.
Bruegge et al. (1998)), GENRA produces an estimate of the *a posteriori* probability distribution function (PDF) for the retrieval
of geophysical parameters. To minimize sensitivity to inherent assumptions, we primarily assess our results comparatively. For
example, we test if it is best to include or exclude viewing geometries at the most extreme angles when working with a radia-
tive transfer operating under the assumption of a plane parallel atmosphere. We should also note that the methods we use to
calculate an *a posteriori* PDF from GENRA are conceptually very similar to Bayesian inference applied to real measurements,
which will be the subject of future work.

This investigation was performed for a retrieval of aerosol and ocean surface properties using only the MISR Near Infrared
(NIR) channel, centered at 865nm. This choice was made because the deep ocean is black (or nearly so) at that wavelength. The
result is a considerable simplification of radiative transfer and the number of parameters needed for remote sensing retrieval.
Ultimately, we intend for our MISR retrieval algorithm to serve as a means to atmospherically correct MODIS observations,
which have spectral sensitivity appropriate for the determination of ocean body properties. To that end, we must establish the
means to accurately determine the atmospheric and ocean surface state with the MISR NIR channel. In a Bayesian context, the
*a posteriori* PDF's of aerosol and ocean surface properties from MISR become the *a priori* PDF's for the MODIS retrieval.

This manuscript is organized as follows. Section 2 is the methodology, covering details of radiative transfer, uncertainty, the
GENRA technique, and the test cases we used for assessment. The results are in Section 3, where we describe the outcome of
our tests. The conclusion (Section 4) contains an evaluation of the meaning of this work and its implications for future study.



## 2 Methodology


Our goal is to build a method to assess the level of parameterization appropriate for simultaneous retrieval of aerosol and ocean surface properties using MISR 865nm multi-angle observations over the ocean. First, we created a LUT representative of such observations at a variety of aerosol and ocean surface conditions under different combinations of solar and observation geometry. Next, we calculated expected measurement and model uncertainty for all elements of the LUT. We then implemented

the GENRA Bayesian information content assessment technique with the goal of understanding the following questions:

– Can accurate retrieval be performed given our choice of parameters?

– How do expectations of retrieval uncertainty vary with solar and observation geometry?

– How do expectations of retrieval uncertainty vary with parameter value?

– Since the underlying radiative transfer assumes a plane parallel atmosphere, is there any benefit to excluding the two
most extreme view angle MISR observations, where plane parallel radiative transfer may be less accurate?

– Is sun glint sufficiently parameterized with a scalar wind speed value, or do we need to parameterize wind in terms of both magnitude and direction?

### 2.1 Radiative Transfer

Radiative transfer calculations were performed using software built to use the method of Ahmad and Fraser (1982). AFRT
treats the model atmosphere as consisting of plane parallel layers, which are homogeneous horizontally, but inhomogeneous vertically. It rests on an optically smooth ocean surface or a rough ocean surface where the slope-orientation of the capillary waves follows Cox and Munk's probability distribution Cox and Munk (1954). In addition, the model atmosphere is non-emitting and consists of standard gas, aerosols and absorbing gases like ozone (although in this implementation we do not test sensitivity to such gases). It accounts for all orders of scattering in the atmosphere, and outputs the four stokes parameters of
the diffused radiation leaving the top and bottom of the atmosphere. Also, it accounts for Fresnel reflection from the ocean surface, but neglects the scattering and absorption in the ocean itself. In a bench mark studies against other radiative transfer calculations (Dave (1972), Fraser and Walker (1968)), the relative accuracy of AFRT for a Rayleigh scattering only atmosphere is 0.15%.

AFRT is used to generate atmospheric correction LUT's in operational use by the NASA Ocean Biology Processing Group
(OBPG) for processing of data from instruments such as the Sea-viewing Wide Field-of-view Sensor (SeaWiFS), MODIS, the Visible and Infrared Imager/Radiometer Suite (VIIRS) and others (see Mobley et al. (2016) for a detailed tutorial). As mentioned previously, contributions from the ocean body beneath the air water interface are not included, but this is a reasonable approximation for the open ocean at the NIR channel for which we are performing our retrieval (865nm) (e.g. Gordon and Wang (1994)).





Aerosols are modeled in AFRT as described in Ahmad et al. (2010), by treating them as bi-modal combinations of fine and coarse sized particles. The coarse size mode is optically spherical and minimally absorbing (consistent with sea salt), while the size and real refractive index of both modes are defined by water uptake associated with eight relative humidity values between 30% and 95%. While this relative humidity may or may not be the same as ambient scene relative humidity, it is a convenient means to represent most of the range of microphysical properties encountered over the ocean. The exception is non-spherical

dust aerosols (Kalashnikova et al. (2008, 2013)), which are encountered at some locations over the ocean and we will address in a future work. These eight fine mode aerosols are combined with the coarse mode at ten different fractions, for a total of eighty aerosol models. These fractions are expressed as the relative volume concentration of the fine mode to the total particle volume concentration, with values of 0%, 1%, 2%, 5%, 10%, 20%, 30%, 50%, 80% or 95%. Table 1 lists the optical properties of these aerosol models.

| Size mode | Relative humidity | Refractive index | Mode radius | Sigma radius |
|---|---|---|---|---|
| fine | 30% | 1.518 - i0.00993 | 0.085 | 0.437 |
| fine | 50% | 1.511 - i0.00956 | 0.086 | 0.437 |
| fine | 70% | 1.489 - i0.00842 | 0.089 | 0.437 |
| fine | 75% | 1.467 - i0.00724 | 0.094 | 0.437 |
| fine | 80% | 1.426 - i0.00511 | 0.105 | 0.437 |
| fine | 85% | 1.404 - i0.00392 | 0.115 | 0.437 |
| fine | 90% | 1.388 - i0.00308 | 0.125 | 0.437 |
| fine | 95% | 1.372 - i0.00226 | 0.138 | 0.437 |
| coarse | 30% | 1.476 - i0.00000 | 0.567 | 0.672 |
| coarse | 50% | 1.469 - i0.00000 | 0.575 | 0.672 |
| coarse | 70% | 1.414 - i0.00000 | 0.679 | 0.672 |
| coarse | 75% | 1.380 - i0.00000 | 0.808 | 0.672 |
| coarse | 80% | 1.363 - i0.00000 | 0.921 | 0.672 |
| coarse | 85% | 1.357 - i0.00000 | 0.985 | 0.672 |
| coarse | 90% | 1.350 - i0.00000 | 1.076 | 0.672 |
| coarse | 95% | 1.341 - i0.00000 | 1.288 | 0.672 |

**Table 1.** Optical properties for the aerosol models used in radiative transfer calculations. Note that refractive indices are specified for 865nm. In the radiative transfer calculations, the fine and coarse mode microphysical properties are combined at one of ten different volume concentrations, where the fine mode contributes 0%, 1%, 2%, 5%, 10%, 20%, 30%, 50%, 80% or 95%. Mode radius is specified in microns.

In AFRT, the ocean roughness is characterized by wind speed, which in turn is related to the Cox and Munk's slope probability distribution of the capillary waves on ocean surface (Cox and Munk (1954)). Not included in the computations are shadowing, and the multiple reflection effects between wave facets. It should be noted that the reflections of both the direct and





diffused radiation at base of the atmosphere (that is, Fresnel reflection properly weighted by the Cox-Munk's slope probability distribution), are properly accounted for.

AFRT was used to calculate top of atmosphere (TOA) radiance at a variety of solar and observation geometries, at a density sufficient such that interpolation to a specific scene geometry has minimal error. Each combination of view zenith angle, $\theta_v$ (the observation vector's angle from nadir), solar zenith angle, $\theta_s$ (the solar illumination vector's angle from nadir), and relative azimuth angle, $\phi$ (the solar minus observation azimuth angles), are simulated. This means that there are 7,744 ($22\times22\times16$) observations simulated for each geophysical state. For analysis, these geometries are interpolated to provide a simulated ob-

servation for a specified MISR geometry (see Section 2.3 for a description). Also, note that in our implementation of AFRT azimuthal symmetry was preserved, so simulations from $\phi = 0°$ to $\phi = 180°$ mirror those from $\phi = 360°$ to $\phi = 180°$.

AFRT simulations were varied across four dimensions of parameter space. These dimensions, and the values over which the simulations were performed, are described in Table 2. A total of 3,600 ($8\times10\times9\times5$) geophysical states were simulated for each of the geometries.

Relative humidity ($r$), as described above, modifies the size distribution and complex refractive index of the aerosols to parameterize water update as described in Table 1. Aerosols are treated as bimodal, lognormal, distributions of fine and coarse size aerosols, represented in volume space as

$$
\begin{aligned}
\frac{dV(r)}{d\ln r} = {} & \frac{V_f}{\sqrt{2\pi}\sigma_f} \exp\left[-\left(\frac{\ln r - \ln r_f}{\sqrt{2}\sigma_f}\right)^2\right] \\
& + \frac{V_c}{\sqrt{2\pi}\sigma_c} \exp\left[-\left(\frac{\ln r - \ln r_c}{\sqrt{2}\sigma_c}\right)^2\right]
\end{aligned}
\tag{1}
$$

where $V$ is volume, $r$ particle radius, $V_f$ and $V_c$ are the volumes of the fine and coarse size mode particles, $r_f$ and $r_c$

their geometric mean radius (in microns) and $\sigma_f$ and $\sigma_c$ their geometric standard deviation. Thus, the fine size mode fraction parameter is $f = \frac{V_f}{V_f+V_c}$. Aerosol optical depth ($\tau$) is defined at 865nm, while the wind speed parameterizes the distribution of specular reflection from the sun by the ocean's surface, as described above.

| Parameter | # | Values |
|---|---|---|
| $r$ | 8 | 30, 50, 70, 75, 80, 85, 90, 95 |
| $f$ | 10 | 0%, 1%, 2%, 5%, 10%, 20%, 30%, 50%, 80%, 95% |
| $\tau$ | 9 | 0.00, 0.05, 0.10, 0.15, 0.20, 0.25, 0.30, 0.40, 0.50 |
| $w$ | 5 | 0.0, 1.87, 4.21, 7.49, 11.70 |

**Table 2.** Geophysical parameters for which AFRT simulations were performed, for a total of ($8\times10\times9\times5$) combinations. $r$ is the relative humidity in percent, $f$ is the aerosol fine size mode volume fraction, $\tau$ is the aerosol optical depth at 865nm, and $w$ is the wind speed in m/s.



Other geophysical parameters were held constant for all simulations, implying expectations of minimal impact on parameterization capability. These included trace gas absorption (for which we are minimally sensitive in the MISR NIR channel, and

can be accounted for in operational processing), total atmospheric pressure (which can also be accounted for in operational processing), and the ocean body contribution as mentioned previously. Surface reflectance contribution due to sea foam was not included in our simulations, but a subsequent analysis found that its inclusion in select cases had a negligible impact on our results.

## 2.2 MISR characteristics

There is a rich literature describing MISR's technical characteristics (e.g. Bruegge et al. (1998, 2002, 2004); Diner et al. (1998)) and retrieval algorithms (e.g. Diner et al. (2005); Kahn et al. (2005); Witek et al. (2019); Garay et al. (2020)), so our description is limited to brief details relevant to this study.

The NASA Terra spacecraft, for which MISR is one of the instruments, is in an orbit with an altitude of 705km and inclination angle of 98.2°, and ground track repeat every sixteen days. It has nine push-broom cameras that are oriented along the satellite track direction, with nominal angles with respect to the Earth surface of 0°, ±26.1°, ±45.6°, ±60.0° and ±70.5°. In the

terminology of MISR data users, cameras are denoted [Df, Cf, Bf, Af, An, Aa, Ba, Ca, Da] in order from the most forward camera to the most aft; roughly seven minutes pass from the first image of a ground location to the last. Each camera has four spectral channels, with center wavelengths at 446.6nm (blue), 557.5nm (green), 671.7nm (red) and 866.4nm (NIR, note we approximate this channel at 865nm). Although there is some variation in camera and channel swath width and inherent spatial resolution, the common resolution to which MISR data are typically analyzed is 1.1km. In this paper, we assume all

camera observations have the same spatial resolution, and that systematic uncertainties such as out-of-band instrument response (Bruegge et al. (2004)) have been corrected.

As mentioned previously, we are assessing the retrieval capability utilizing only the NIR channel. By doing so, we reduce the dimensionality of our parameter retrieval space by making the assumption that the water body does not contribute to the

observations. This assumption has a long history in the ocean color remote sensing community (e.g. Gordon and Wang (1994)) and is appropriate for most of the open ocean. It is also assumed to be the case for both the red and NIR channels in the most recent operational MISR aerosol retrieval algorithm, V23 (Garay et al. (2020)). Treatment of turbid or coastal water bodies would require a more extensive parameter space that we will not address in this work.

By using all NIR multi-angle observations, including those made at geometries with reflected sun glint, we can retrieve a

somewhat unique set of parameters: those describing the nature of that sun glint and the atmospheric aerosols above it. The ultimate goal is to aid the atmospheric correction for ocean color observations for the other instrument on Terra, MODIS. Standard atmospheric correction for that instrument has access to single view observations at two of NIR channels and also utilizes the dark ocean assumption. While future missions may make use of a greater number of channels (Ibrahim et al. (2019)), this means that only two pieces of information are used to select an aerosol type (model) and magnitude ($\tau$). With the MISR

NIR multi-angle observations, however, we have nine pieces of information from which to identify the aerosol magnitude ($\tau$),





two parameters defining aerosol type ($r$ and $f$) and a measure for the distribution of reflected sun glint ($w$). How well we expect to retrieve these parameters, given measurement and model uncertainty, is the goal of this work.

## 2.3 Test cases

Retrievals of aerosol properties with multi-angle instruments such as MISR are sensitive to the specific observation geometry (e.g. Knobelspiesse et al. (2012); Knobelspiesse and Nag (2018)), which varies with location, orbit, and season. In order to span the range of potential geometries, we selected seven scenes from the MISR archive. Specifically, we used the SeaWiFS Bio-optical Archive and Storage System (SeaBASS) (https://seabass.gsfc.nasa.gov/, Werdell et al. (2003)) to identify cloud free coincident observations by MISR and ground instruments from AERONET-OC (Zibordi et al. (2010)). AERONET-OC is a network of ground based instruments that are often mounted on platforms at sea and measure both aerosol properties and ocean reflectance. In a future work, we will compare MISR retrievals from the algorithm we develop to the optical properties observed by AERONET-OC, and consider the differences between them in the context of this ICA.

|   | Site | $\theta_s$ | Latitude | Longitude | Date | Time | $\tau(869nm)$ |
|---|------|-----------|----------|-----------|------|------|---------------|
| **A** | COVE-SEAPRISM | 30.5° | 36.90°N | 75.71°W | 2008-04-17 | 15:13Z | 0.052 |
| **B** | MVCO | 27.9° | 41.33°N | 70.57°W | 2008-05-05 | 15:00Z | 0.149 |
| **C** | Helsinki-Lighthouse | 44.3° | 59.95°N | 24.92°E | 2009-08-06 | 09:03Z | 0.056 |
| **D** | Venise | 69.7° | 45.31°N | 12.51°E | 2011-01-03 | 09:34Z | 0.024 |
| **E** | Venise | 69.8° | 45.31°N | 12.51°E | 2011-12-28 | 09:40Z | 0.045 |
| **F** | USC-SEAPRISM | 19.6° | 33.56°N | 118.12°W | 2012-05-27 | 18:00Z | 0.078 |
| **G** | MVCO | 43.9° | 41.33°N | 70.57°W | 2013-09-24 | 15:01Z | 0.010 |

**Table 3.** Location and time of selected AERONET-OC sites with coincident MISR observations. The solar and observation geometries of these sites were used in our ICA. More details on the individual sites can be found at https://aeronet.gsfc.nasa.gov/.

Table 3 contains the details about the seven geometries we use for this study, while Figure 1 is a polar plot of the corresponding MISR observation geometries. The seven cases were chosen to span the range of measurement geometries, and include two at 'high' $\theta_s$ ( 70°), two at 'medium' $\theta_s$ ( 45°) and two at 'low' $\theta_s$ ( 30°). We also included a less common extremely low $\theta_s$ ( 20°) for which the glint was centered about the An (nadir) viewing direction. Among these geometries, the 'high' $\theta_s$ scenes are minimally impacted by glint, while its influence progressively increases for lower $\theta_s$.

## 2.4 GENRA

There are multiple techniques for assessing information content in a remote sensing retrieval. Inherent to all techniques is the need to connect measurement space to geophysical parameter (often denoted state) space, in a manner the incorporates measurement and model characteristics plus *a priori* knowledge. While such efforts can not incorporate 'unknown unknowns,' they provide a useful ceiling for potential retrieval success and a means to compare different measurement and retrieval systems.





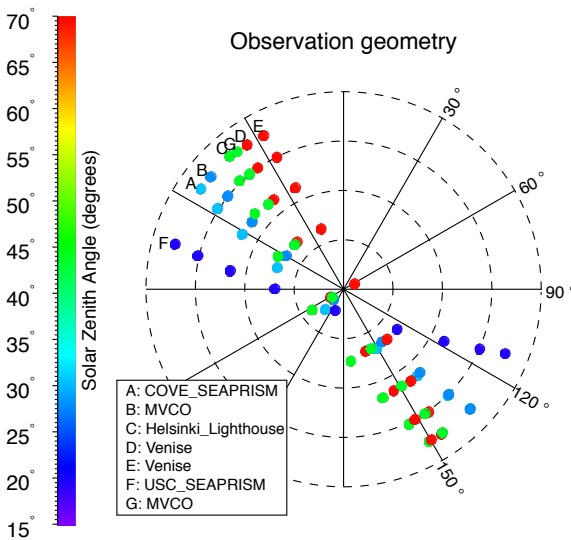

**Figure 1.** Polar plot of the observation geometries used in the ICA, corresponding to AERONET-OC and MISR coincident observations. The angular coordinate indicates $\phi$, while the radial coordinate indicates $\theta_v$ (each dashed concentric circle represents $20°$). $\theta_s$ is indicated by color.

The aerosol remote sensing community has often utilized a technique popularized by Rodgers (2000), which projects model and measurement uncertainty from the measurement to parameter spaces. This technique is fast and convenient, as it uses Jacobian matricies (**K**) calculated as the partial derivative of the measurements with respect to retrieval parameters. This is

performed using a radiative transfer forward model ($F$), which represents geophysical reality by calculating a simulated observation (**y**) for a given set of parameters (**m**), i.e. $\mathbf{y} = F(\mathbf{m})$ and $K_{i,j} = \frac{\partial F_i(\mathbf{m})}{\partial \mathbf{m}_j}$ (where $i$ and $j$ are indices for the measurement and parameter vectors, respectively). Like all information content assessments, it relies on the suitability of the forward model and uncertainty estimate fidelity. Because of its speed and flexibility, it has become a common tool in the multi-angle aerosol remote sensing to identify optimal instrument characteristics (e.g. Hasekamp and Landgraf (2005), Lebsock et al. (2007), Wa-

quet et al. (2009), Knobelspiesse et al. (2012), Ottaviani et al. (2013), Xu and Wang (2015), Knobelspiesse and Nag (2018), and Hasekamp et al. (2019)). Furthermore, since Jacobians are often used in optimal estimation or similar iterative retrieval algorithms, this technique can also reuse the Jacobians of the final iterative step to provide an estimate of parameter retrieval uncertainty (e.g. Knobelspiesse et al. (2011a, b)). It has also been shown to produce similar results to other assessment techniques, such as in Gao et al. (2020).

However, the Rodgers (2000) technique does make several assumptions that may influence the analysis. The use of Jacobians implies that the forward model connecting parameter to measurement space is locally linear. Furthermore, PDF's are assumed to be Gaussian, as uncertainty is characterized by a simple distribution width metric (although an advantage of that technique is the simplicity with which it accounts for uncertainty correlation for multiple measurements). For information rich, well posed,





problems this is most likely not an issue, as measurement uncertainties are well characterized by Gaussian distributions and
because *a posteriori* PDFs are usually sufficiently compact that the local linearity assumed in the use of Jacobians is preserved.
Additionally, computational efficiency is important in the case of large measurement and parameter space dimensionality. For
example, in the assessment of the Aerosol Polarimetry Sensor (APS) in Knobelspiesse et al. (2012), the measurement vector
had 3,570 elements for the retrieval of 12 parameters. This contrasts with the current study, which has a measurement vector
with 9 elements for the retrieval of 4 parameters.

The smaller dimensionality of this retrieval allows us to explore other techniques that do not make the same approximations
as Rodgers (2000). The GENRA technique (Vukicevic et al. (2010)) is attractive because it does not make the same assumptions of forward model linearity and the Gaussian nature of uncertainty. Developed for the assessment of cloud remote sensing
(Coddington et al. (2012); Coddington et al. (2013); Coddington et al. (2017), GENRA is similar to Bayesian inference in its
treatment of all components of the retrieval system as stochastic with associated PDFs. The result is a posterior which repre-
sents the best understanding of the expected parameter PDF for a synthetic measurement. The technique is computationally
inexpensive and can utilize pre-computed LUT's to represent the forward model (the primary expense is the typical need to interpolate these LUT's to a finer grid). Metrics for the reduction in entropy from the prior to posterior PDF, such as the Shannon
information content (Rodgers (2000)) can be used to characterize the overall quality of a retrieval, while marginal PDFs (the
posterior PDF integrated to one parameter dimension) indicate the capability for an individual parameter.

GENRA can be represented by a single, perhaps deceptively simple, equation:

$$p_o(\mathbf{m}) = \frac{p_r(\mathbf{m})}{\gamma} \sum_{\mathbf{y}} p_d(\mathbf{y}) p_l(F(\mathbf{m})|\mathbf{m}) \qquad (2)$$

where $p_o(\mathbf{m})$ is a multi-dimensional *a pOsteriori* PDF for the $\mathbf{m}$ parameter vector (bold indicates vector), $p_r(\mathbf{m})$ is the *a pRiori* PDF, $p_d(\mathbf{y})$ is the stochastic measurement distribution (d for data), and $p_l(\mathbf{y})$ is the same for the Likelihood function.
Instead of an integration over measurement space ($\mathbf{y}$), we use a summation, as we represent all PDF's as discrete functions. $\gamma$
is a normalization factor such that

$$\gamma = \sum_{\mathbf{m}} p_o(\mathbf{m}) \qquad (3)$$

which ensures the summation of the posterior PDF is one. The result is a multidimensional *a posteriori* PDF that incorpo-
rates all known information. In the context of GENRA the 'data' ($p_d$) is a PDF created from a single node in the LUT and
expectations of measurement and model uncertainty, while the likelihood ($p_l$) is created from the entire LUT, which is used
as a forward model ($\mathbf{y} = F(\mathbf{m})$). In our implementation, the *a priori* PDF ($p_r(\mathbf{m})$) is a weakly informative prior defined as
uniform within the boundaries of the LUT and zero outsize those boundaries.

A full assessment of retrieval space involves the calculation of the *a posteriori* PDF for each LUT node. The result will have
$n + n$ dimensions, where $n$ is the number of dimensions in $\mathbf{m}$. In our case, this is a rather unwieldy eight dimensions. Marginal
PDF's ($p_m$) ease this analysis by calculating the summation of $p_o(\mathbf{m})$ over each retrieval parameter:





$$p_m^r(r', r, f, \tau, w) = \sum_f \sum_\tau \sum_w p_o(\mathbf{m}) \tag{4}$$

$$p_m^f(f', r, f, \tau, w) = \sum_r \sum_\tau \sum_w p_o(\mathbf{m}) \tag{5}$$

$$p_m^\tau(\tau', r, f, \tau, w) = \sum_r \sum_f \sum_w p_o(\mathbf{m}) \tag{6}$$

$$p_m^w(w', r, f, \tau, w) = \sum_r \sum_f \sum_\tau p_o(\mathbf{m}) \tag{7}$$

where we have now reduced the total number of dimensions to five, meaning a marginal PDF (with the associated dimension
indicated by $'$) for each parameter in each node of the LUT.

A useful metric to indicate the overall success of a measurement is the Shannon information content, $SIC$ (Shannon and
Weaver (1949)). Derived from the concepts of entropy, $SIC$ indicates the reduction in the volume of possible solutions in
$p_o(\mathbf{m})$ from the original $p_r(\mathbf{m})$, and is calculated:

$$S_{posterior} = -\sum_L p_o(\mathbf{m}) \log_2 p_o(\mathbf{m}) \tag{8}$$

$$S_{prior} = -\sum_L p_r(\mathbf{m}) \log_2 p_r(\mathbf{m}) \tag{9}$$

$$SIC = S_{prior} - S_{posterior} \tag{10}$$

$$SIC_H = (S_{prior} - S_{posterior})/S_{prior} \tag{11}$$

Here, $S$ represents entropy and we have performed summations of discrete PDF's. Like in Coddington et al. (2012) and
Shannon and Weaver (1949), we choose to use a base 2 logarithm so that the entropy values are represented in bits. *SIC* is the
change in that entropy between *prior* and *posterior*. It is bounded by zero, which indicates a no information in a measurement,
and an upper value which is $\log_2(k)$, where $k$ is the number of bins in the discrete PDF. *SIC* can be calculated for the entire
multi-dimensional space (using $p_o(\mathbf{m})$ and $p_r(\mathbf{m})$), or for specific parameters using the marginal PDF's and the corresponding
subset of $p_r$. However, we will not be able to compare *SIC* since the number of bins in the discrete PDF $k$ varies among
parameters. For that purpose, we define a relative Shannon information content, $SIC_H$, for which the maximum is one.

## 2.5 Uncertainty

Our ICA incorporates expectations of both measurement and model uncertainty. The former are based upon the MISR instru-
ment characteristics, while the latter are derived from our knowledge of radiative transfer model accuracy and estimates for
uncertainty due to simplifications of that model compared to geophysical reality. We test the relative importance of those sim-
plifications compared to other sources of uncertainty as part of this work. Furthermore, appropriate estimates of measurement



Atmospheric
Measurement
Techniques

Discussions

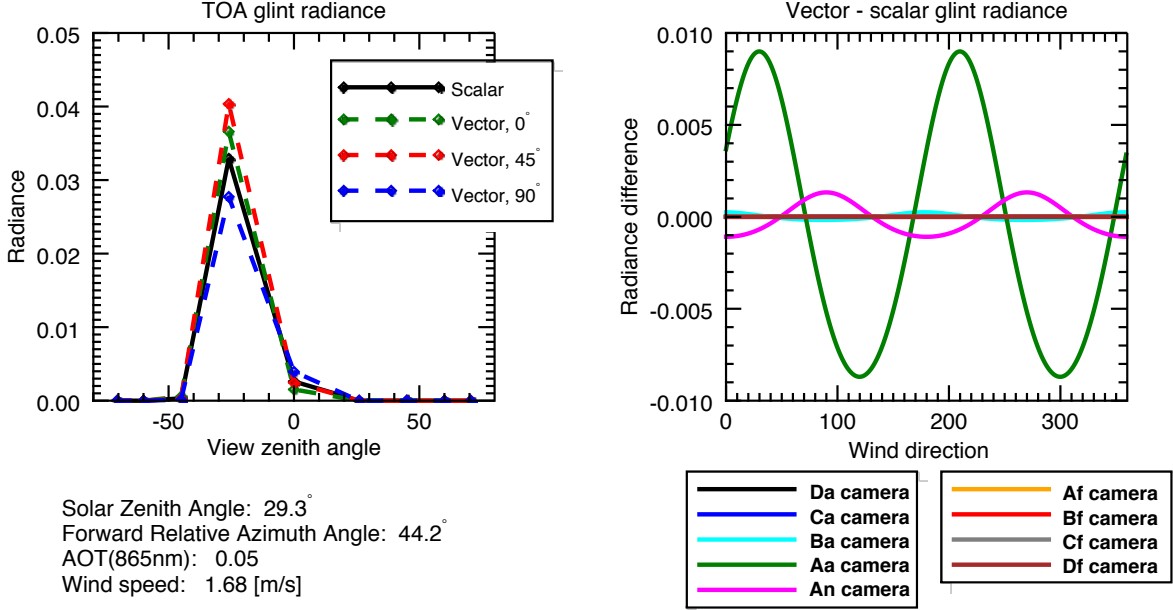

**Figure 2.** Differences between scalar (unidirectional) and vector (magnitude and direction) treatment of wind influenced sun glint. The left panel represents the top of atmosphere (TOA) contribution to radiance due to ocean surface sun glint for a scene similar to test case A. The scalar parameterization is shown in black, while green, red and blue dashed lines represent glint represented for the same wind speed (1.68m/s) but different headings. The right panel shows the scalar - vector difference in terms of TOA radiance for each MISR camera in the same scene, as a function of wind direction. Note that the Aa camera view, at $26°$ aft, is closest to the reflected sun glint peak at $29.3°$ and thus shows the largest impact of wind direction. These differences are used to create an uncertainty estimate, $\sigma_{cm}$, that represents the potential error for scalar wind speed glint parameterization compared to that of the vector model.

and model uncertainty provide for the means to assess retrieval performance with varying observation geometries and retrieval parameter values.

Additional sources of error ('unknown unknowns') can be due to conditions outside of the simulated parameter LUT boundaries (e.g. $\tau >> 0.5$) or optically relevant complexity (such as non-spherical aerosols described in Kalashnikova et al. (2008), to be addressed in a future work) that are not incorporated into the analysis because we are unable to estimate their specific impact in either measurement or parameter space. In this sense, the ICA can be considered the best case scenario, as additional sources of error can only degrade performance.

Lacking more detailed information about our uncertainties, we treat them as Gaussian and construct measurement PDF's ($p_d$) as follows

$$p_d(y) = \frac{1}{\sigma\sqrt{2\pi}} \exp\left(\frac{(y-y')^2}{-2\sigma^2}\right) \tag{12}$$





where $y'$ is the simulated LUT value and $\sigma$ represents the squared sum of all sources of uncertainty. In this representation we express the $p_d$ for each measurement (i.e. each camera observation) individually.

Measurement uncertainties for MISR are based upon the comprehensive literature that has characterized various sources of uncertainty (e.g. Bruegge et al. (1998, 2002, 2004); Garay et al. (2020); Witek et al. (2018, 2019)) from which we chose the following model of MISR radiometric uncertainty

$$\sigma_{MISR} = 0.03 \max(y', \frac{0.04 \cos\theta_s}{\pi}) \qquad (13)$$

where $y'$ is expressed in units of radiance. For all of our simulations, the LUT radiance was calculated such that the exo-atmospheric irradiance is $1 \ W/m^2$ and the solar distance in astronomical units is 1. While this convention is not the same as geophysical reality, it is easily converted to the appropriate values if needed and irrelevant for the purposes of this study so long as we are consistent. The relationship between reflectance ($\rho$) and radiance ($y'$) is thus

$$\rho = \frac{\pi y'}{\cos\theta_s} \qquad (14)$$

so the right-most portion of equation 13 is a conversion from reflectance units to the version of radiance in use in this work. In most cases, this was the smaller of the pair of values in that equation, and not incorporated into the uncertainty estimate.

We consider three sources of model uncertainty. The first is the AFRT numerical uncertainty established by benchmark comparisons to be $\sigma_{RT} = 0.001y'$ (Ahmad and Fraser (1982)). Next, we include expectations of uncertainty due to the atmospheric

plane-parallel assumption inherent to AFRT. While this assumption has minimal impact for most solar and viewing geometries, it becomes relevant at oblique angles, such as the MISR Df and Da cameras. Recently, Frouin et al. (2019) conducted an analysis of the difference between plane-parallel and pseudo-spherical radiative transfer. We use this as the basis for an estimate of the plane parallel model uncertainty ($\sigma_{pp}$), described in Table 4. Later, we will use this camera specific uncertainty estimate to test if the information content of a retrieval is higher when using all nine cameras, or when omitting the two most oblique

angle cameras (Df and Da).

| Camera | $\theta_v$ | $\sigma_{pp}$ |
|--------|------------|---------------|
| Df, Da | $\pm70.5°$ | $0.0041y'$ |
| Cf, Ca | $\pm60.0°$ | $0.0011y'$ |
| Bf, Ba | $\pm45.6°$ | $0.0002y'$ |
| Af, Aa, An | $\leq |26.1|°$ | $0.0$ |

**Table 4.** Uncertainty estimate due to the plane parallel radiative transfer assumption, based upon analysis by Frouin et al. (2019).

AFRT uses a scalar (unidirectional) treatment of sun glint with a single wind speed parameter. Because this represents a potential oversimplification by the forward model, we include this as an additional model uncertainty term. The magnitude of this uncertainty depends on geometry, wind speed and atmospheric transmittance, so it is derived individually for each





measurement by comparing the scalar and vector (based on wind speed and direction) sun glint models of Cox and Munk
(1954). Figure 2 illustrates the differences between the scalar and vector parameterizations as they would be observed by
MISR for a viewing geometry similar to test case A, a wind speed of 1.68m/s and $\tau = 0.05$. The left panel shows how the
greatest differences between the scalar and vector representations of sun glint occur at the peak angle of that glint, while the
right panel shows sinusoidal nature of the scalar and vector difference. To determine the uncertainty term, $\sigma_{cm}$, we calculate
the cumulative distribution function of the differences between scalar and vector glint radiance at the top of atmosphere for a
full 360° cycle. We then assume the differences are Gaussian distributed and estimate $\sigma_{cm}$ based on the shape of the cumulative
distribution function. This conservative assessment was chosen because we do not account for the wave shadow effect, although
we expect that to be small for MISR geometries (Saunders (1967)).

## 2.6 Implementation

To summarize, we use radiative transfer simulations to create a LUT, which we interpolate to each of the geometries described
in 2.3. This LUT is again interpolated to a finer parameter grid and used to generate the Likelihood function. It is also used,
in combination with uncertainty estimates, to determine individual measurement distributions for each node in the LUT. In
practice, this involves the following steps:

1. generate LUT$(\theta_v, \theta_s, \phi, r, f, \tau, w)$ using AFRT

2. interpolate LUT to specific test case geometry: LUT$_{MISR}(c, r, f, \tau, w)$, where $c$ represents the MISR camera observa-
tions

   (a) interpolate LUT$_{MISR}$ to a fine parameter grid for the likelihood function: LUT$_{big}(c, r', f', \tau', w')$ where $'$ indicates
       interpolated parameter values

   (b) for each $(r, f, \tau, w)$ node in LUT$_{MISR}$ (indicated by $i$):

      i. create the *a priori* PDF, $p_r$, as uniform within the LUT parameter bounds
ii. step through each camera view, $c_j$, and do the following:

         A. determine the specific simulated radiance $y' = $ LUT$_{MISR}(c_j, r_i, f_i, \tau_i, w_i)$
         B. calculate and add all the uncertainties in quadrature: $\sigma^2 = \sigma_{MISR}^2 + \sigma_{RT}^2 + \sigma_{pp}^2 + \sigma_{cm}^2$
         C. construct the measurement PDF, $p_d(y)$, as in equation 12 using $y'$ and $\sigma$
         D. create the likelihood function, $p_l$, using LUT$_{big}$
E. combine $p_r$, $p_d$, and $p_l$ as in equation 2 to determine the *a posteriori* PDF, $p_o$
         F. update the *a priori* PDF such that $p_{r,j} = p_o$
      iii. based on $p_o$, calculate SIC$_R$ for this LUT node, plus the corresponding marginal PDF's and their SIC$_H$ values

The product of this analysis is a multidimensional *a posteriori* PDF, $p_o(r, f, \tau, w, r', f', \tau', w')$, where the first four dimen-
sions index each node in the LUT, while the latter four represent the interpolated parameter values. An overall estimate of





Shannon Information Content has four dimensions, $SIC_R(r, f, \tau, w)$, and there are corresponding Shannon Information Content values derived from the marginal PDF's have the same dimensionality. These products are generated for each test case geometry, so comparisons among them can be used to determine the impact of changes in geometry. Slight modifications to this procedure can be used to address specific questions. For example, we test the importance of scalar wind/glint parameterization by omitting the $\sigma_{cm}$ term from the $\sigma$ calculation in step B. We also test the value of discarding the most oblique

view angles, and their corresponding increased model uncertainty due to the plane parallel approximation, by simply omitting those measurements in the loop in step ii. The multi-dimensional nature of these products is a challenge we hope to address successfully in the next section.

## 3   Results

### 3.1   Overall assessment

The final product of our analysis is an *a posteriori* PDF ($p_o$) that has eight dimensions. There are individual PDF's calculated for each test case geometry. As this is difficult to visualize, we calculate aggregate metrics such as the $\text{SIC}_H$ and marginal PDF's. Figure 3 is an example uncertainty assessment for test case G, parameter set $r = 90\%$, $f = 0.10$, $\tau = 0.10$, and $w = 1.87 m/s$. In this geometry, the influence of the sun glint is obvious (panel A), so we would expect to be able to retrieve all parameters. Indeed, this is the case: marginal PDF's (panels I, J, K and L) are narrow and peaked near the simulated parameter value.

Slices through $p_o$ (panels C, D, E, F, G, and H) show a small volume centered about the simulated parameter values. $\tau$ and $w$ have the narrowest marginal PDF's, while those of the aerosol intensive parameters $r$ and $f$ are wider. What is striking is that the shape of the marginal PDF's is often non-Gaussian, especially for the wider PDF's. This indicates that single parameter retrieval uncertainty estimates based upon Gaussian assumptions may not adequately represent the nature of that uncertainty. To some extent, this is to be expected for the wider PDF's, as they are more influenced by the uniform prior.

Figure 4 shows another slice of $p_o$ for test case G. The parameter set is $r = 75\%$, $f = 0.50$, $\tau = 0.30$, and $w = 7.49 m/s$, and overall performance is worse than in Figure 3. The larger $\tau$ obscured the sun glint feature, leading to reduced retrieval capability for all parameters (among other changes). This is shown in the wider marginal PDF's, which are clearly non-Gaussian, the larger volumes in the *a posteriori* PDF slices, and the lower $\text{SIC}_H$ values. The change in $\text{SIC}_H$ helps us understand relative differences between the two cases. In this case, the overall metric for $\text{SIC}_H$ decreases to 0.35 from 0.54, and all four of the

marginal $\text{SIC}_H$ values decrease. This is especially true for $w$, which decreases to 0.27 from 0.82.

   Next, we consider the results for a different test case. Figure 5 is for test case D, where $\theta_s = 69.7°$, and the same parameter set as in Figure 3. At this geometry, sun glint is not present in the simulated observations. The overall impact is a degradation in $w$ retrieval capability, as expected, and improvement in the aerosol parameters. Despite the lack of sun glint in the simulation, the marginal PDF for $w$ does correctly exclude large wind speeds, which perhaps might have influenced the observations. In

any case, the total $\text{SIC}_H$ decreases only slightly, to 0.48.

   While these three figures only show results for three test case and parameter set examples, they express the range of potential results, and the relationship between *a posteriori* PDF volumes, marginal PDF's and $\text{SIC}_H$. In the following subsections we

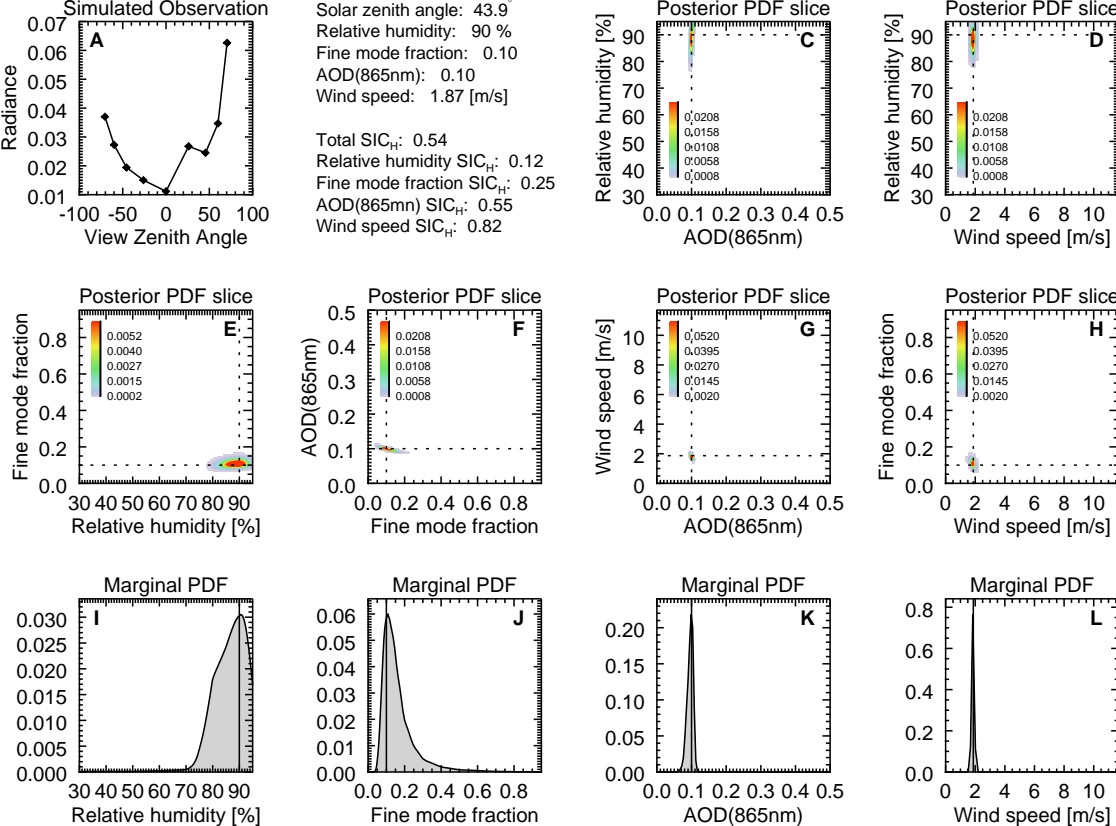

**Figure 3.** GENRA results for a successful retrieval, whose overall SIC$_R$ = 0.54. The geometry for this scene is for test case G, which had a $\theta_s = 43.9°$, with $r = 90\%$, $f = 0.10$, $\tau = 0.10$, and $w = 1.87 m/s$. Marginal PDF's for each of those parameters are shown in panels I, J, K and L, respectively, and solid vertical lines in those plots indicate simulated parameter value. Slices through *a posteriori* space are in panels C, D, E, F, G, and H, where dashed lines indicate the value of the LUT node under investigation. Panel A contains the simulated observation.

will focus on the change in information content with geometry, parameter value, and sensitivity to model assumptions. To do so, we only plot SIC$_H$, which is the most compact means to express information content. However, figures similar to those in this section were made for all test cases and parameter sets, and can be found in the archive at [to be made available prior to publication].

### 3.2 Sensitivity to geometry

Multi-angle observations, especially those that contain sun glint, should be very sensitive to sun and observation geometry (e.g. Knobelspiesse et al. (2012); Ottaviani et al. (2013)). This is because the atmospheric radiative transfer is largely governed by

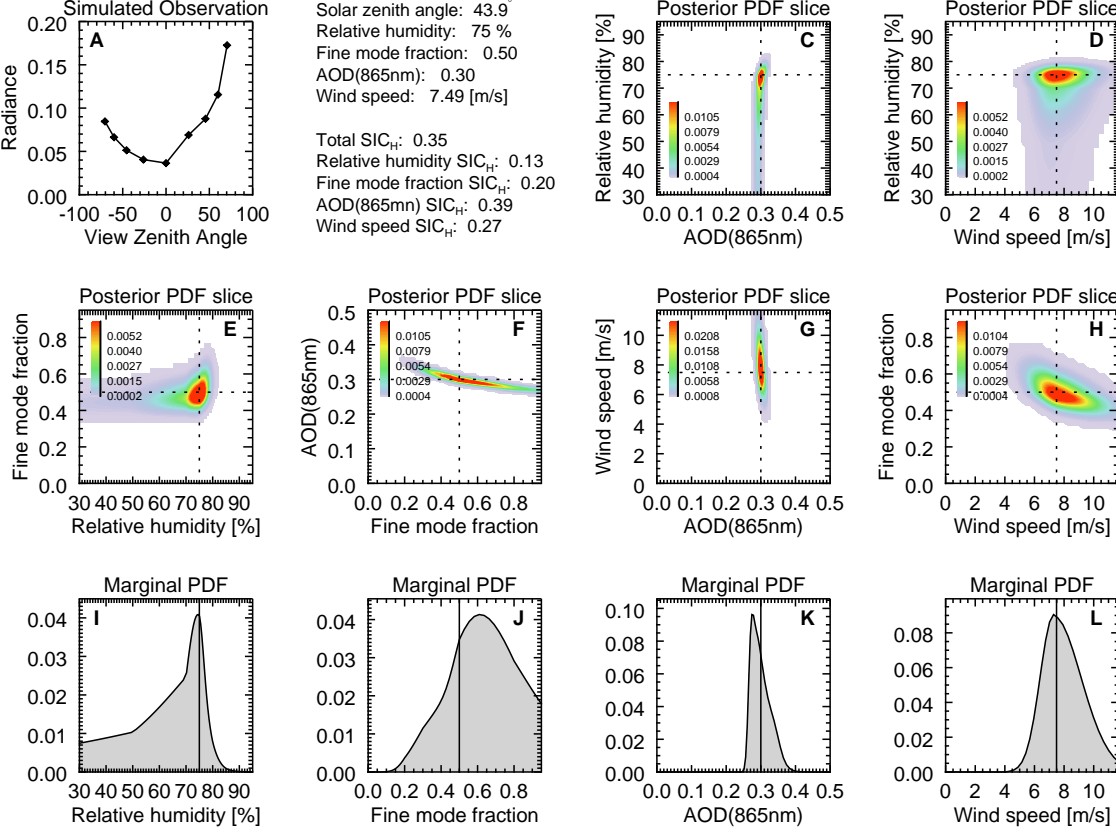

**Figure 4.** GENRA results for a less successful retrieval, whose overall $SIC_R = 0.35$. The geometry for this scene is also for test case G, which had a $\theta_s = 43.9°$, while these parameters are $r = 75\%$, $f = 0.50$, $\tau = 0.30$, and $w = 7.49 m/s$. Marginal PDF's for each of those parameters are shown in panels I, J, K and L, respectively. Slices through *a posteriori* space are in panels C, D, E, F, G, and H, where dashed lines indicate the value of the LUT node under investigation. Panel A contains the simulated observation.

the scattering angle (the angle between illumination and observation vectors, Hovenier (1969)), while the glint is controlled

by both sun and observation geometry (Cox and Munk (1954)). This means that what is observed by MISR's nine cameras

changes with location, time of day, and season, factors which define geometry. It also means that the nature of the sun glint

pattern does not change in tandem with that of atmospheric scattering.

As we can see from Figure 1, MISR cameras observe a scene in a generally constant relative azimuth angle plane, indicated

by straight lines in that figure. While the significance of such behavior is explored more fully in Knobelspiesse and Nag

(2018), we would like to note that if this plane is aligned with the solar principal plane, i.e. $\phi = 0°, 180°$, then a multi-angle

measurement would see the widest possible range of scattering angles, presumably containing more information about the



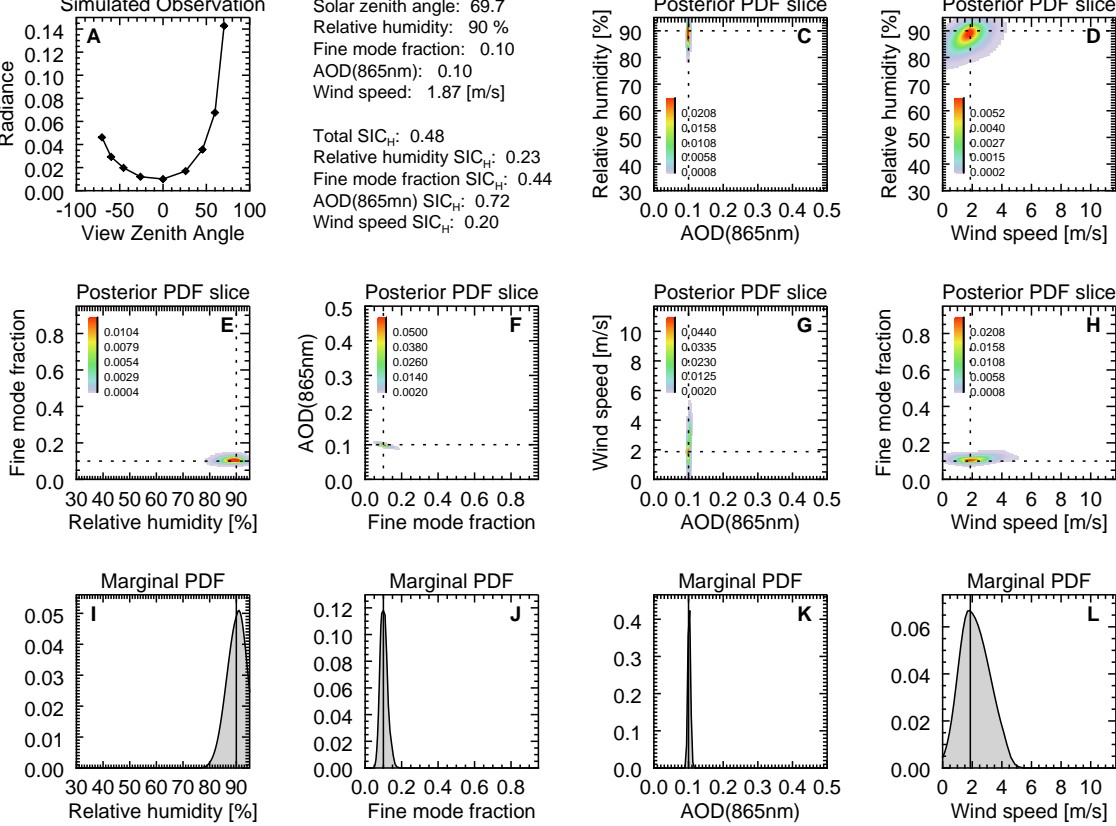

**Figure 5.** GENRA results for a retrieval with the same parameters as in Figure 3, but for test case D. Now the solar zenith angle is $\theta_s = 69.7°$, so the sun glint is not observed (or at least is not obvious from the simulated observations in panel A). Parameters remain the same, with $r = 90\%$, $f = 0.10$, $\tau = 0.10$, and $w = 1.87 m/s$, while overall $\text{SIC}_H$ decreases slightly to 0.48. As expected, $\text{SIC}_H$ for $w$ decreases significantly, while it increases for all the other (aerosol) parameters.

atmospheric state. It would also be more influenced by reflected sun glint, which is centered about that plane. Observations along the cross principal plane ($\phi = 90°, 270°$), in contrast, see a narrower range of scattering angles, and are less likely to

observe sun glint (which is also defined by $\theta_s$). MISR observations fall between these two extremes. The test cases with the largest $\theta_s$ (cases D and E) are closest to the solar principal plane, meaning that they observe the largest scattering angle range. However, $\theta_s$ is high enough that the sun glint is for the most part not observed. With MISR, $\theta_s$ and $\phi$ mostly change in tandem, so as $\theta_s$ decreases, the observation plane becomes closer to the cross principal plane. For these observation (A, B and F), a narrower scattering range, yet greater glint influence, is to be expected.

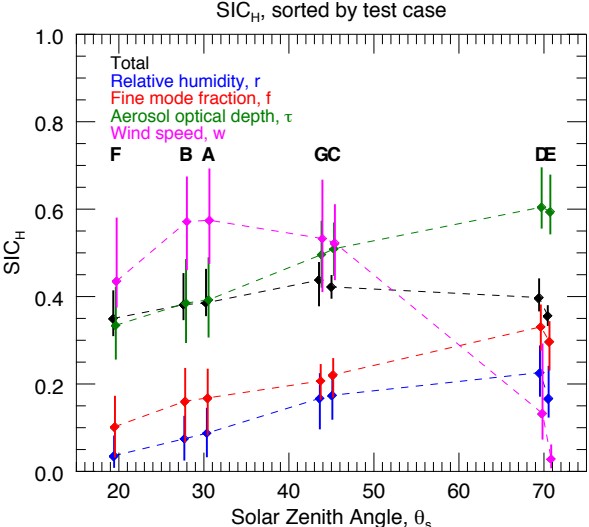

**Figure 6.** $SIC_H$ sensitivity to sun and observation geometry. Median $SIC_H$ values for all cases with the specified Solar Zenith Angle ($\theta_s$) are plotted with diamonds connected by dashed lines. Vertical bars indicate the inter-quartile range. The total $SIC_H$ is in black, while the marginal $SIC_H$ for $r$, $f$, $\tau$ and $w$ are plotted in blue, red, green and magenta, respectively. Letters showing test case geometry are also shown. To aid legibility, in some cases we offset the marginal $SIC_H$ $\theta_s$ values slightly (less than 1 degree).

Figure 6 shows the impact of different observation geometries on $SIC_H$. In this figure, we have plotted the median $SIC_H$ for each test case / geometry as diamonds connected by dashed lines. Vertical lines are the inter-quartile $SIC_H$ range. This has been done for the total $SIC_H$, shown in black, along with the marginal $SIC_H$ for relative humidity (blue), fine mode fraction (red), aerosol optical depth (green) and wind speed (magenta). The overall $SIC_H$ is relatively insensitive to geometry, perhaps because $SIC_H$ improvement for some parameters comes at the expense of others. For the aerosol parameters, $SIC_H$ increases

with $\theta_s$ (and consequently, a decrease in $\phi$), confirming our expectation that access to a greater scattering angle range improves the information content. By contrast, wind speed, which is the parameter retrieved from the sun glint pattern, shows the highest $SIC_H$ for moderate $\theta_s$ (test cases A, B, C and G), and nearly zero for the largest $\theta_s$ (test cases D and E). This is to be expected, since the sun glint has minimal impact on those scenes.

  Test case F is somewhat unique, as it is among the lowest $SIC_H$ for nearly all parameters. While this is due in part to the

most restricted scattering angle range of a nearly cross principal plane geometry, this scene also exhibits a form a symmetry, since the highest glint is in the nadir view (camera An). The other camera angles are mirrored in the forward or aft views, implying a reduction in information content. Figure 7 demonstrates this for the same parameter set as in Figures 3 and 5. In this figure we can see the aforementioned nearly mirror symmetry in panel A. While $w$ and $\tau$ still have narrow marginal PDF's, those for $r$ and $f$ have grown. Panel E indicates that there is the some non-uniqueness between the pair of parameters, which

has a detrimental effect on information content. This is similar to the lowered information content found in some low latitude observations by Kalashnikova et al. (2013).

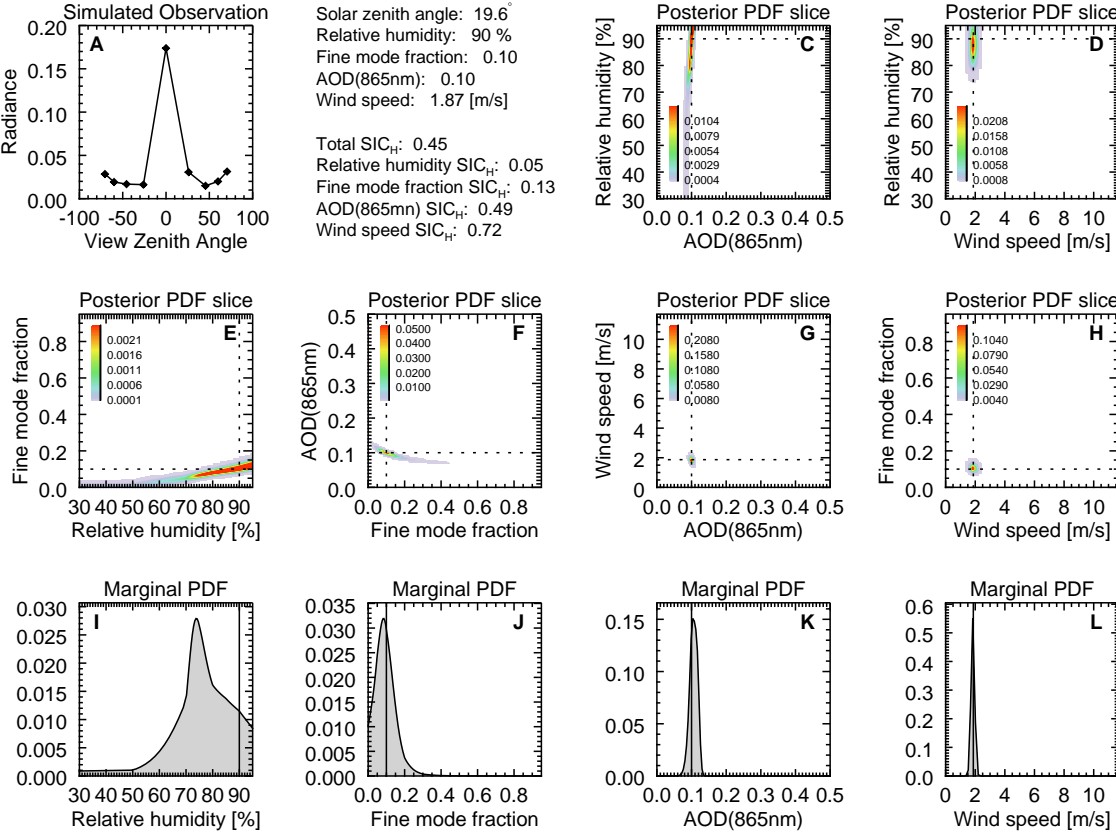

**Figure 7.** GENRA results for a retrieval with the same parameters as in Figure 3, but for test case F. The solar zenith angle is $\theta_s = 19.6°$, so the sun glint centered about the most nadir view camera, An. Parameters remain the same, with $r = 90\%$, $f = 0.10$, $\tau = 0.10$, and $w = 1.87 m/s$, while overall $\text{SIC}_H$ decreases further to 0.45.

## 3.3 Sensitivity to parameter value

In addition to geometry, the retrieval parameters defining a scene also impact information content. For example, for low aerosol loads the information content of aerosol intensive properties (in this parameterization scheme, $r$ and $f$) should be less than that

for higher aerosol loads. However, with highly nonlinear systems such as ours, it can be difficult to make predictions beyond simplistic assumptions such as these. GENRA, however, provides the means to systematically assess the relationship between parameter value and retrieval information content.

Figure 8 shows the sensitivity of both the total and marginal $\text{SIC}_H$ values for each of the retrieved parameters. For clarity, only median values of $\text{SIC}_H$ are presented. For the aerosol intensive parameters ($r$ and $f$, top row) we see only small changes in





SIC$_H$ as those parameters change. This is consistent with Knobelspiesse et al. (2012), who also found little to no information content dependence on aerosol intensive properties (see Figure 4 in that paper). That study was an analysis using different parameterization schemes, radiative transfer and information content assessment technique. This tells us that the range of aerosol intensive property values has minimal impact on information content assessment result. Furthermore, so long as the paramaterization scheme correctly represents geophysical reality, retrieval success will depend more on observation geometry

and other parameters. However, we should note that this assessment does not include non-spherical aerosols, such as dust, for which information content (and required parameterization scheme) differs.

The bottom panels in figure 8 do show SIC$_H$ sensitivity to parameter value. The SIC$_H$ of aerosol intensive properties, as noted above, is lowest for small amounts of aerosols, and increases with aerosol optical depth. This is to be expected, since there will be a greater radiative impact of the aerosol intensive parameters as the quantity of aerosols increases. Conversely,

the ability to determine $w$ is best for low aerosol loads, and decreases as the aerosol magnitude increases and obscures the reflected sun glint. SIC$_H$ for $\tau$ has a more complicated relationship with the value of $\tau$ itself, in part because this is an extensive parameter. The total SIC$_H$ shows relatively small change with $\tau$, indicating a balance between aerosol intensive and $w$ information content with changing $\tau$. In contrast, SIC$_H$ uniformly decreases for all parameters as $w$ increases. Increasing $w$ serves to broaden the sun glint pattern, which may be a source of information content reducing ambiguity.

## 3.4 Sensitivity to the plane parallel assumption

In Section 2.5, we discuss the uncertainty associated with the use of a (computationally efficient) radiative transfer model that simulates the atmosphere as parallel layers. Frouin et al. (2019) assessed that uncertainty, which we incorporate into our overall uncertainty estimate independently for each MISR camera (Table 4). As we can see, the Df and Da cameras, whose $\theta_v = \pm 70.5°$, have the largest uncertainty, roughly an order of magnitude less than the overall instrumental uncertainty

(Equation 13).

This raises the question: is it better to omit those two camera views when performing retrievals with plane parallel radiative transfer? To test this, we compare our overall results, described in the previous subsections, to those for a modified GENRA test. The modified test does not incorporate simulated observations from the Df and Da cameras, such that the iteration described in step (b)iii in Section 2.6 does not include those views. In other words, we are testing if cameras Df and Da, when including

expectations of their uncertainty due to the plane parallel assumption, contain information that does not already exist in the other views.

We find that it is indeed best to include all nine MISR cameras, so long as the expectations of the plane parallel model uncertainty are a part of the retrieval algorithm. Figure 9 illustrates the differences between the SIC$_H$ for the case using all nine MISR cameras, and for the seven camera case excluding the most extreme view zenith angles. The top panel is the

histogram of the SIC$_{H,9views}$−SIC$_{H,7views}$ difference. Positive values indicate that the SIC$_H$ is higher for the case using all camera than when the Df and Da camera views are omitted. The vast majority are positive, and in some instances significantly so. This is true for both the overall SIC$_H$ and the parameter specific marginal SIC$_H$ values. In an effort to understand the consistency of these differences, we plotted in the lower panel the median SIC$_H$ for each test case. While these are all positive,



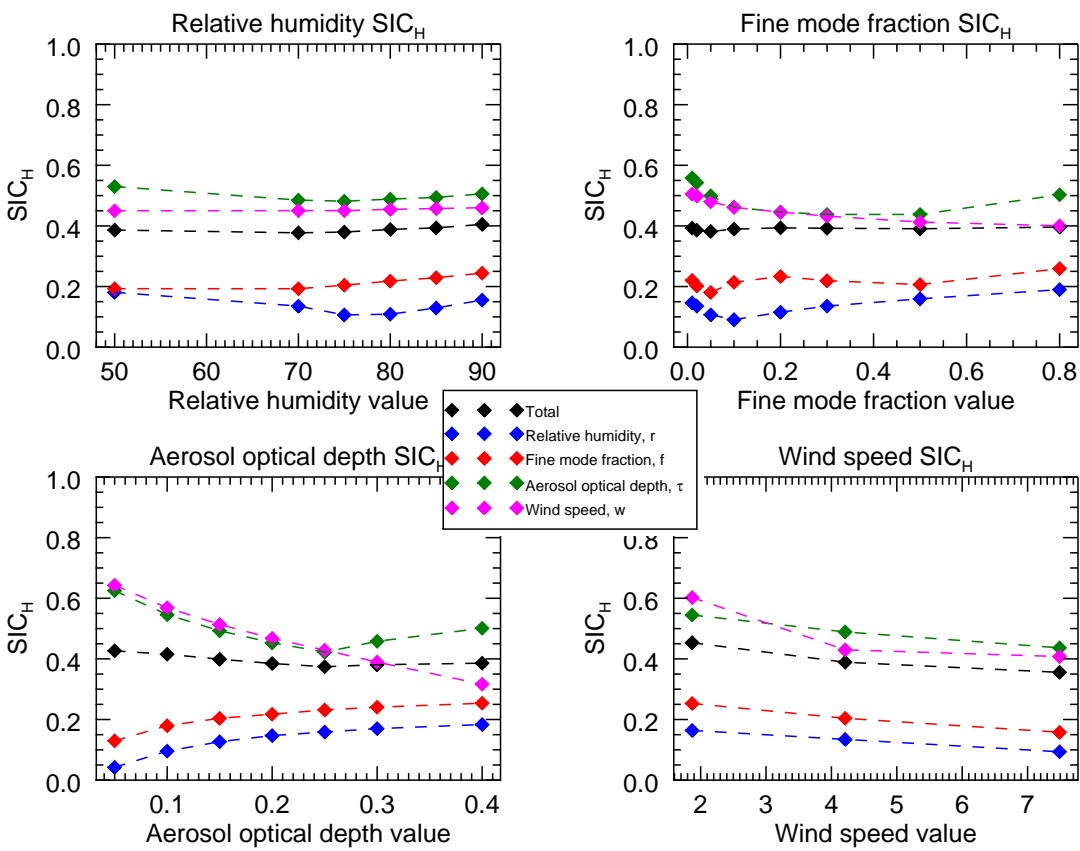

**Figure 8.** Median $SIC_H$ for total (black), $r$ (blue), $f$ (red), $\tau$ (green) and $w$ (magenta) values, parsed by parameter value. The top left figure shows sensitivity to $r$, top right to $f$, bottom left to $\tau$ and bottom right to $w$.

the $SIC_H$ for test case C is higher than the others. This is unexpected because test case G has a similar $\theta_s$ but less positive

$SIC_{H,9views} - SIC_{H,7views}$ values. If we refer to Figure 1, we see that the plane of observations in test case C is farther from nadir than that of test case G. This means that the sun glint is observed at higher $\theta_v$ angles, so the omission of the most extreme $\theta_v$ cameras has a more significant impact upon information content. This also illustrates that the geometry differences between the nadir portion of the image swath and points far from that direction are can in some cases be significant.

### 3.5    Sensitivity to the sun glint assumption

In a similar fashion, we test sensitivity to the parameterization model of reflected sun glint. As described in Section 2.5, Cox and Munk (1954) provided two sun glint parameterization models. The simplest, which we use, treats wind as scalar and unidirectional. A more complicated model uses wind speed and direction (or the equivalent zonal and meridonal winds). The

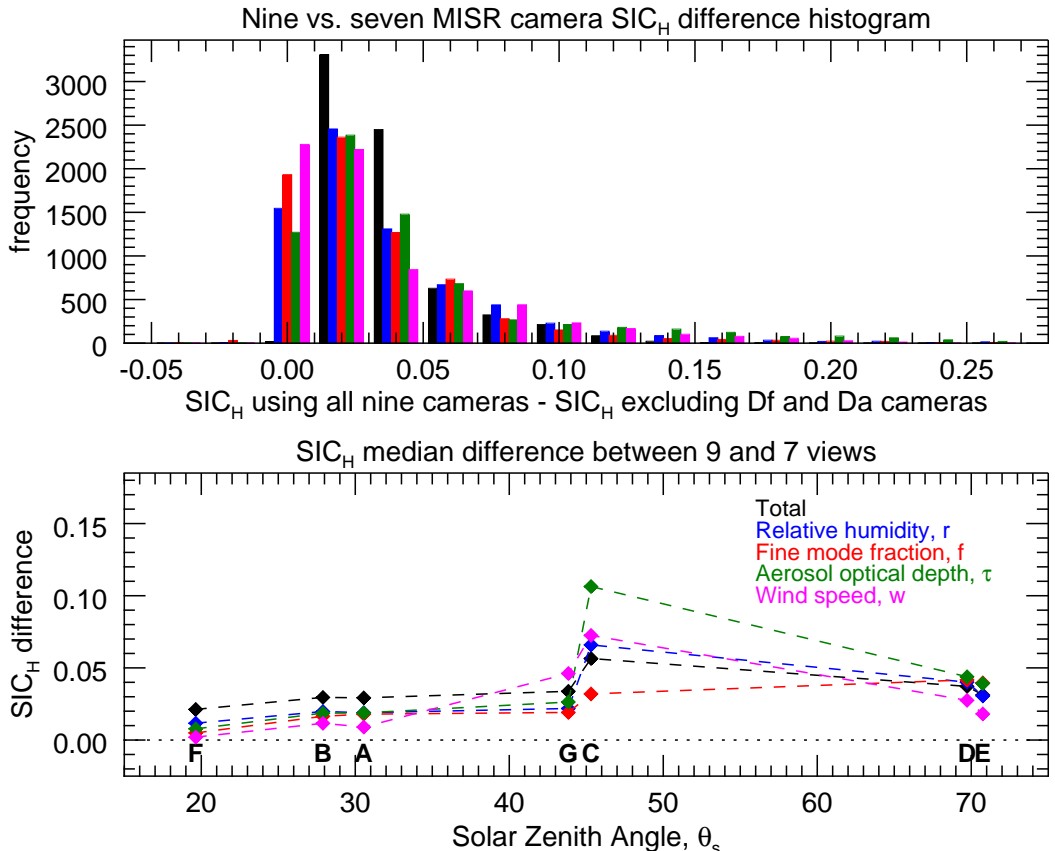

**Figure 9.** Test of the influence of the plane parallel radiative transfer approximation. An additional GENRA information content analysis was performed without the Df and Da MISR cameras, which have the most extreme $\theta_v$ values and are thus subject to the most plane parallel radiative transfer errors. The top panel contains the histograms of the original $SIC_H$ minus the $SIC_H$ for the excluded Df and Da. Universally positive values mean that Df and Da do contain useful information, despite their additional error. The lower panel is the median $SIC_H$ for each of the test cases. These too are positive. Test case C is especially positive, perhaps due to the off-nadir nature of that scene's geometry.

latter is difficult to use in our current implementation of AFRT because that model assumes azimuthal symmetry ($180° < \phi < 360°$ is a mirror reflection of $0° < \phi < 180°$). It also implies the retrieval of two, not one, parameters.

We do account for the additional model uncertainty due to the use of scalar instead of the vector representation of wind speed, and the results thus far include that uncertainty. To test its significance, we performed another GENRA analysis without that source of uncertainty, and compare to our original test case. The results are shown in Figure 10. Like Figure 9, this shows the histogram of the $SIC_H$ change, in this case without and with the scalar uncertainty term. For most cases, but not all, the information content difference is minimal. The lower panel, which has median $SIC_H$ values broken down by test case



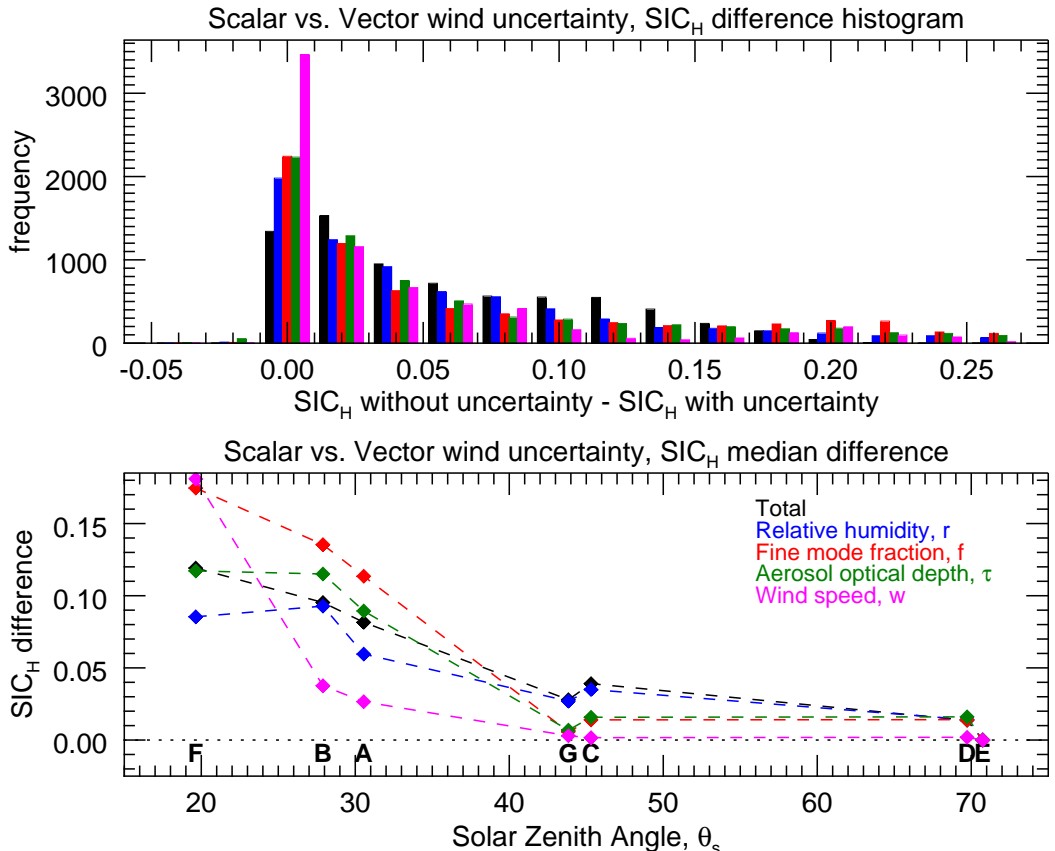

**Figure 10.** Test of the influence of the scalar versus vector treatment of wind speeds. Our parameterization scheme uses a scalar representation of wind speed, and the information content assessment incorporates an estimate of the model uncertainty due to this approach compared to the more complex vector winds. This figures shows the $SIC_H$ difference when using, or not using, that model uncertainty. The top panel contains the histograms of the $SIC_H$ without vector wind uncertainty minus the $SIC_H$ when that has been included. Universally positive values mean that vector wind model uncertainty does have an impact on the results. The lower panel is the median $SIC_H$ for each of the test cases. Here we see that the lowest $\theta_s$ cases, where the influence of sun glint is greater, are most sensitive to this model uncertainty.

geometry, shows that the lowest $\theta_s$ test cases are most affected by this form of uncertainty, which makes sense since they are most influenced by sun glint.

From a practical perspective, how significant are these impacts? While $SIC_H$ is a useful differentiation metric, examination of the marginal PDF's can help place them in the context of retrieval success. Figure 11 shows the change in the marginal PDF for each parameter for the set of parameters illustrated in Figures 3, 5 and 4 ($r = 90\%$, $f = 0.10$, $\tau = 0.10$, and $w = 1.87 m/s$).

In this figure, dashed lines indicate the marginal PDF without the vector wind uncertainty; lack of visible dashed lines means





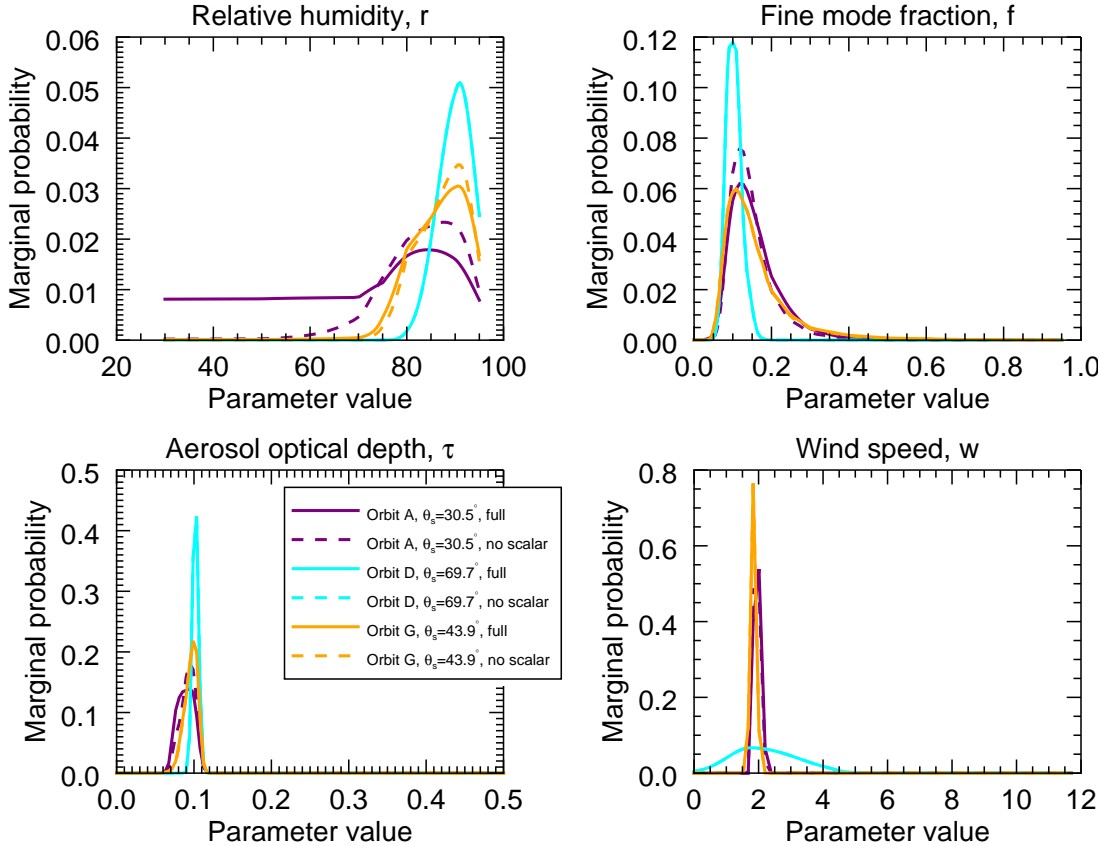

**Figure 11.** Marginal PDF's for the parameter sets in Figures 3, 5 and 4 ($r = 90\%$, $f = 0.10$, $\tau = 0.10$, and $w = 1.87 m/s$). Solid lines indicate assessment with the full uncertainty model, including model error estimates for the use of a scalar rather than vector sun glint parameterization. Dashed lines are the marginal PDF's for the case where that term has been omitted. Lack of dashed lines for a case indicate that there is no difference in the marginal PDF compared to that using the full model uncertainty.

they overlap the solid line PDF's representing use of the vector wind uncertainty. As we can expect based on Figure 10, the lowest $\theta_s$ cases are more impacted by this uncertainty term. However, the impacts appear constrained primarily to the aerosol intensive parameters ($r$ and $f$) with minimal impact on $\tau$ and no impact on $w$. This indicates that determination of wind speed is largely unaffected by the use of the wind parameterization scheme, while the scalar scheme only degrades the ability to
determine aerosol intensive parameters moderately and at low $\theta_s$ geometries.





## 4 Conclusions

The primary purpose of this research is to establish that multi-angle measurements from the MISR 865nm channel are sufficient to determine the combined aerosol and sun glint state. Using a radiative transfer model in active use by the NASA Ocean Biology Processing Group, (AFRT) and the GENRA information content assessment technique, we have demonstrated that this is the case. This is shown for set of four parameters: the scalar (unidirectional) wind speed ($w$), aerosol optical depth at 865nm ($\tau$) and two aerosol intensive parameters (relative humidity, $r$, and fine size mode fraction, $f$), which define the aerosol size distribution and refractive index.

Some of our most important results include the following:

1. The available information content for each parameter in a MISR 865nm observation varies with geometry. The solar and observation geometry defined by the MISR orbit includes high solar zenith angles close to the solar principal plane, while the relative azimuth angle progressively increases as solar zenith angle decreases at low latitudes. Low solar zenith angle observations are affected by the reflected sun glint, and as we see in Figure 6, and have high $\mathrm{SIC}_H$ for the retrieval of $w$. As the solar zenith angle increases, so does the $\mathrm{SIC}_H$ for the aerosol parameters ($r$, $f$, and $\tau$), while that of $w$ decreases. The increase is due to the greater observed range of scattering angles. The overall $\mathrm{SIC}_H$, however, remains relatively constant.

2. Marginal PDF's can not always be represented with a Gaussian numerical distribution. While this can be expected for parameter sets close to the boundaries of the uniform *a priori* PDF's, it is also encountered far from those boundaries, such as in panel A of Figure 4. It appears that as SIC increases and the marginal PDF narrows, marginal PDF's become more Gaussian. This indicates that single parameter uncertainty estimates produced by a retrieval algorithm may not sufficiently represent true uncertainty, especially in the case of less well retrieved parameters.

3. In most cases, $\mathrm{SIC}_H$ does not depend much upon retrieved parameter values. This means that retrieval performance should be consistent regardless of parameter value. The exception is $\tau$: as it is increased, $\mathrm{SIC}_H$ increases for aerosol intensive properties while decreasing for $w$. Additionally, there small reduction in $\mathrm{SIC}_H$ for all parameters as $w$ increases.

4. Despite higher model uncertainties due to the plane parallel assumption, the Df and Da (largest view zenith angle) observations should always be used in a retrieval (assuming these errors do not impose a systematic bias).

5. The use of a scalar wind model for sun glint is a source of model uncertainty compared to the more detailed vector wind model. However, this impact only occurs for the lowest solar zenith angle scenes, and in those cases only slightly increases the marginal PDF's of the aerosol intensive parameters. For these reasons we do not believe the added complication of using two parameters (which also requires a different radiative transfer code) is necessary for these observations.

As in all information content assessment using synthetic observations, these results are only as realistic as the parameterization scheme and estimates of measurement and model uncertainty. Some uncertainties are difficult to quantify, including



those associated with an inappropriate parameterization, i.e. scenes not included in the simulated LUT. For this reason, this assessment can be considered a best case scenario: we know the information content can not be any better than this, and it
could be worse if uncertainty is underestimated.

Two types of scenes that are not included in our LUT are those that have water leaving radiance at 865nm, and those that have non-spherical aerosols such as dust. This is in large part because those aspects of the radiative transfer were not easily implemented in the current version of AFRT. We are in the process of incorporating the spheroid models of Dubovik et al. (2006) into AFRT. Future studies will also look into simulations with a simple parameter to characterize water leaving
radiance, which at 865nm is most likely due to suspended sediment, shown to have an isotropic Bidirectional Reflectance Distribution Function (BRDF) (Hlang et al. (2012)). Because of these limitations, we expect this study to be valid for global oceans unaffected by highly turbid coastal regions or areas of continental dust outflow over the ocean.

While the GENRA technique is intended for information content assessment, the approach can be easily modified to be a Bayesian inference style retrieval for real observations. Here, we have calculated *a posterior* PDF's for each node of the LUT.
A retrieval algorithm would instead operate on real observations. Minor modifications would need to be made to the LUT to account for changes in total atmospheric pressure, the influence of trace gases, and the non-physical radiometric units we used in this work (see Section 2.5). The largest limitation is the need to interpolate the LUT to a fine grid spacing, which would probably be too computationally expensive for operational processing with the brute force method we used. However, there are a number of computational techniques available to address this issue, such as the Markov chain Monte Carlo class of
algorithms for sampling a PDF.

The next step in our algorithm development process is to generate a corresponding LUT for MODIS wavelengths. This will provide the means to connect the observed aerosol and glint state determined with MISR observations to their radiometric impacts at the MODIS wavelengths used in subsequent Ocean Color algorithms. That knowledge will be used to atmospherically correct TOA MODIS observations.

Finally, we should say that the GENRA information content assessment technique can also be applied to existing algorithms that make use of LUT's. This could be a powerful evaluation tool for historical retrievals.

*Code and data availability.* Code is available upon request. Individual figures similar to Figure 3, for all test cases and parameters, are to be made externally available upon publication.

*Author contributions.* KK secured funding for and leads this project, performed the analysis, and wrote the manuscript. ZA and SA created
the AFRT simulations with infrastructure support by BF, SB and JG. AI, RL, ZA and MG provided input to the manuscript, and all provided guidance throughout.





*Competing interests.* The authors declare no competing interests.

*Acknowledgements.* This work was funded by NASA program NNH17ZDA001N-TASNPP, "The Science of Terra, Aqua, and Suomi-NPP,"
managed at NASA Headquarters by Dr. Laura Lorenzoni. We would also like to thank the MISR instrument team for their help, and Dr.
Ralph Kahn, NASA Goddard Space Flight Center and Dr. Odele Coddington, University of Colorado, LASP, for fruitful discussions.



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
