# Peer review of "Analysis of simultaneous aerosol and ocean glint retrieval using multi-angle observations"

_Atmospheric Measurement Techniques, 2020_

## Referee Comment (RC1) · Anonymous Referee #2 · 20 Dec 2020

The manuscript describes in great details a theoretical analysis of the information content attached to the MISR satellite instrument in one given spectral band (i.e., centered on 865 nm) but performing acquisitions for nine distinct viewing directions. In this part of the spectrum, most of the ocean might be considered as virtually totally absorbing that is to say that the water leaving radiance is nil. Even if the assumption is a little restrictive (e.g., intense bloom), it can be advantageously used to get purely atmosphere and water surface information. Here, the authors discussed how accurate could be achieved retrieval of some key parameters concerning aerosols and air-water interface roughness given the nine pieces of information provided by the MISR directional measurements. This information content assessment is performed upon a sophisticated Bayesian approach and outcomes of a well-established radiative transfer code. The

results obtained for a limited set of "test cases" show not very surprising results: for low aerosol optical thickness (AOT), surface parameters are better retrieved and when AOT increases the aerosol model is better retrieved.

The manuscript is well-written with a sound mathematical background for such analysis. However, the parameters used in the analysis could be expanded to better delineate the optimal number of parameters to be estimated. More importantly, the primary goal of the analysis is not very clear and should be specified; is the study dedicated to: (i) estimation of aerosol microphysical parameters, (ii) atmospheric correction for ocean color purposes, (iii) sea surface roughness characterization (or (iv) all on the same time). For the first case, the study should include more aerosol parameters to be tested (single scattering albedo, mean radius and variance of the modal size distribution...). For (ii), the most important parameter is the spectral variation of the atmospheric radiance. As to (iii), the surface model should be furthered with inclusion of foam formation, for instance, and discussed in light of the uncertainties attached to wind-sea-roughness model with the isotropic and directional implementation (see (Breon Henriot, 2006; Munk, 2009)) and compare with other technical approaches (see (Harmel Chami, 2013)). In any case, the representativeness of the parameters retrieved from the near-infrared band should be analyzed over the visible-NIR spectral range. The study could conclude on the benefits of using the methods developed for the "aerosol" algorithms to the "atmospheric correction" ones, and respectively.

Minor comments:

The study is presented based on a few "test cases" corresponding to some AERONET-OC cases. First, those sites are mostly coastal with non-null NIR water-leaving radiance. Second, for such a theoretical study there is no need to restrict the analysis to very few and too specific conditions. For the sake of completeness, this test cases should be removed and replaced with a complete set of configurations, for instance sun angle from $0°$ to $90°$, aerosol optical thickness from 0 to 1, wind speed from 0 to 12 m/s (of course, actual values are at the discretion of the authors).

Technical comments:

Through the manuscript: remove statement on future works, this gives the impression that everything is still to be done.

Title: specify the main purpose: atmospheric correction, aerosol retrieval. . . (see major comments)

L.16: "virtually black"

L.125: it would be very interesting to include more complex aerosol models than those obtained based on Mie assumptions (non-spherical, heterogeneous. . .)

Table 1: specify the distribution type (in number, surface or volume) for the modal parameters

L.138. Provide the values of the increments used

L.156: foam should be considered but if not you have to remove wind speed greater than a certain threshold (8, 10 or 12 m/s)

Section 2.3: to be removed

L.199: "a less common extremely low $\theta_s(20°)$", why is it less common, actually we can have sun zenith angle = $0°$ for subtropical acquisitions.

L.209: I would say: "the partial derivatives of the simulated signal in the vicinity of the retrieved parameters"

L.211: in equation with $\delta m_j, m_j$ shouldn't be bold.

L.220: PDF not defined

L. 221: "measurement space is locally linear"? "locally continuous", instead?

L.253: it is not very clear to me, why eight dimensions?

Table 4: $\sigma_{pp}$ is also a function of sun zenith angle

**References**

Breon, F. M., Henriot, N. (2006). Spaceborne observations of ocean glint reflectance and modeling of wave slope distributions. Journal Of Geophysical Research-Oceans, 111, C06005.

Harmel, T., Chami, M. (2013). Estimation of the sun glint radiance field from optical satellite imagery over open ocean: multidirectional approach and polarization aspects. Journal of Geophysical Research, 118(1), 1–15. https://doi.org/10.1029/2012JC008221

Munk, W. (2009). An Inconvenient Sea Truth: Spread, Steepness, and Skewness of Surface Slopes. Annual Review of Marine Science, 1(1), 377–415. https://doi.org/10.1146/annurev.marine.010908.163940

---

## Referee Comment (RC2) · Hongqing Liu (Referee) · 2 Jan 2021

This manuscript has an interesting topic of assessing the information content of the MISR multi-angle measurements aiming for a potential simultaneous retrieval of aerosol and wind speed. By using AFRT to generate a lookup table and applying GENRA for ICA, several sensitivity tests are designed to reveal the optimal choices of the retrieval strategy. I think this is a good paper, and I don't have any disagreements regarding the analysis and conclusions; but I do have some questions about the technique, especially how to implement the GENRA approach and perform the sensitivity test.

My general understanding (not sure it is right) of this work is that, instead of using real

[Figure]

MISR data, the simulated MISR measurements for given sets of parameters (r,f,$\tau$,w) at the geometries of the test cases (interpolated from the LUT) would be used as "inputs" to derive the posterior PDF of the parameters (r',f',$\tau$',w'), where the prime indicates the "retrieval space" which is much finer than that of the "input" (as in the LUT), so "the result will have n+n dimensions" (line 253-254).

My first question is for equation 2: what is the "likelihood function"? It is said "Pd(y) is the stochastic measurement distribution (d for data), and Pl(y) is the same for the likelihood function" (line 243). Is the "Pl" has the same function form as "Pd" (equation 12), but using different y'? From Vukicevic's paper (2010), her equation 1 has the stochastic measurement distribution (Pd) and model distribution (Pt) terms, and the "likelihood" is the "combining of the model and measurement pdfs" (section 2.3), which seems to correspond to the "Pd(y)" only, since "the data (Pd) is a PDF created from a single node in the LUT and expectations of measurement and model uncertainty" (line 248-249), as defined in the equation 12, the uncertainty variance ($\sigma$) includes both the measurement and model contributions. In this paper, the "likelihood function" is mentioned at five places, but it is hard to me to get a clear understanding of this term.

My second question is about the last two sensitivity tests (sections 3.4 and 3.5). It appears to me (again, might be wrong) that the simulated MISR measurements (from LUT) are calculated with the plane-parallel and scalar wind speed assumptions. It is a little hard to understand how to test the sensitivity to the conditions whose signals are not in the inputs. For the test of the plane parallel assumption (section 3.4), if the simulated Df and Da camera observations are from the plane parallel calculations, then it would be expected that retrieval with them (9-camera) would have a better result than without (7-camera), since the (extra) information contained in these two cameras are consistent with the retrieval assumptions (both assume plane parallel). For the sensitivity test of the glint, discarding the uncertainty associated with vector wind would only lead to the improvement since the inversion would be more consistent with the input (both assume the scalar wind without penalty). If I didn't misinterpret the result,

figure 10 seems indicate that the test without the scalar uncertainty term would have more information content than the test with the uncertainty since the SIC difference is positive.

Other than these two questions, I think the paper is well written and the analysis is thorough and clear. So I would recommend this paper be accepted after some clarification about the above questions are made.

One minor issue, for the equations 4-7, should the summation be make over the parameters with prime (i.e., r', f' etc.)?

---

## Referee Comment (RC3) · Anonymous Referee #1 · 13 Jan 2021

This study presents a theoretical information content analysis to evaluate the capability of multi-angular MISR 865nm measurements to simultaneously determine aerosol and ocean surface sun glint variables, including AOD, fine-mode fraction, relative humidity, and surface wind speed. Such analysis was accomplished by a radiative transfer simulation of seven sets of MISR observation geometries and the Bayesian-based information content (and error) analysis. Authors also performed several comparative information analysis to evaluate the sensitivity of retrieval errors to solar & view geometries, plane-parallel atmosphere assumption in radiative transfer, and wind direction.

This is an important study, as indicated by the authors, which is to lay the groundwork for the design of an algorithm for a simultaneous retrieval of aerosol and ocean surface properties from MISR. The paper is well written and well organized. I have the following

comments that author may consider:

1. It seems to me that correlation between errors of individual observations and correlation between the prior error of each state elements are not considered in the Bayesian information analysis. However, these error covariances may exist between MISR's multi-angular observations, as well as between some retrieved variables (such as AOD and fine-mode fraction, AOD and RH). It is thus necessary to explicitly declare the assumption of error correlation in this work and discuss its potential impacts to the results.

2. I agree with the reviewer #1 that more analysis should be performed to extend the currently selected geometrical conditions. It will be of interest to see the results for MISR observations in solar principal plane and perpendicular plane (which are currently missing in the selected geometries, as in Figure 1).

3. In the section of results, readers need to compare posterior PDFs for different conditions but tend to get lost to flip back-and-forth between Figures 3 to 7. I would recommend to rearrange the panels in Figure 3-7 by combining same panels for different conditions. Doing so will help the comparative interpretation.

4. The symbol "r" is used to represent both relative humidity and particle radius. Authors may want to use different symbols for them to avoid confusion.

---

## Author Comment (AC3) · 18 Feb 2021

Thank you for your review, our response is attached -KDK

Please also note the supplement to this comment:
https://amt.copernicus.org/preprints/amt-2020-423/amt-2020-423-AC3-supplement.pdf

---

## Author Response (AR1)

**Author response to reviews of "Analysis of simultaneous aerosol and ocean glint retrieval using multi-angle observations" by Kirk Knobelspiesse et al.**

**Feb 8, 2020**

We would like to express our gratitude to the reviewers for their thorough and positive assessment. Below are our individual responses to each reviewer, reviewer's text in italics.

We would also like to note that we have added supplementary material at data.nasa.gov at the following location:
https://data.nasa.gov/Earth-Science/MISR_MODIS_AtmCorrection/sg4r-ftwb
This contains figures similar to 3, 4, 5 and 7, but for each of the 7,000+ simulations at various geometries and parameter states. We have added a note regarding this in the 'code and data availability section.

**Anonymous Referee #2**

*The manuscript describes in great details a theoretical analysis of the information content attached to the MISR satellite instrument in one given spectral band (i.e., centered on 865 nm) but performing acquisitions for nine distinct viewing directions. In this part of the spectrum, most of the ocean might be considered as virtually totally absorbing that is to say that the water leaving radiance is nil. Even if the assumption is a little restrictive (e.g., intense bloom), it can be advantageously used to get purely atmosphere and water surface information. Here, the authors discussed how accurate could be achieved retrieval of some key parameters concerning aerosols and air-water interface roughness given the nine pieces of information provided by the MISR directional measurements. This information content assessment is performed upon a sophisticated Bayesian approach and outcomes of a well-established radiative transfer code. The results obtained for a limited set of "test cases" show not very surprising results: for low aerosol optical thickness (AOT), surface parameters are better retrieved and when AOT increases the aerosol model is better retrieved.*

*The manuscript is well-written with a sound mathematical background for such analysis. However, the parameters used in the analysis could be expanded to better delineate the optimal number of parameters to be estimated. More importantly, the primary goal of the analysis is not very clear and should be specified; is the study dedicated to: (i) estimation of aerosol microphysical parameters, (ii) atmospheric correction for ocean color purposes, (iii) sea surface roughness characterization (or (iv) all on the same time). For the first case, the study should include more aerosol parameters to be tested (single scattering albedo, mean radius and variance of the modal size distribution...). For (ii), the most important parameter is the spectral variation of the atmospheric radiance. As to (iii), the surface model should be furthered with inclusion of foam formation, for instance, and discussed in light of the uncertainties attached to wind-sea-roughness model with the isotropic and directional implementation (see (Breon Henriot, 2006; Munk, 2009)) and compare with other technical approaches (see (Harmel Chami, 2013)). In any case, the representativeness of the parameters retrieved from the near-infrared band should be*

analyzed over the visible-NIR spectral range. The study could conclude on the benefits of using the methods developed for the "aerosol" algorithms to the "atmospheric correction" ones, and respectively.

Our goal is item (ii). The final paragraph of the abstract starts with: "An algorithm designed upon these principles is in development. It will be used to perform an atmospheric correction with MISR for coincident ocean color (OC) observations by the Moderate Resolution Imaging Spectroradiometer (MODIS) instrument, also on the NASA Terra spacecraft."
To further clarify we added the following language to the conclusion (new language in italics)

The primary purpose of this research is to establish that multi-angle measurements from the MISR 865nm channel are sufficient to determine the combined aerosol and sun glint state *for the purposes of AC.*

We also added the following paragraph to the conclusion:
*The intent of this algorithm is to provide for an AC that can be applied to observations at VIS wavelengths. We build upon an approach (Mobley et al., 2016) that has been used for OC remote sensing for decades. This rests on two assumptions. First, as is described above, premise that the ocean body does not contribute in the NIR, and that the aerosols can be treated as spherical. Second, the approach hypothesizes that aerosol refractive index is spectrally invariant within VIS-NIR wavelengths. Of course, this is not always the case, as is reviewed by Frouin et al., 2019. However, with this algorithm, we show improvement over single view angle techniques, including glint/wind speed sensitivity and a better ability to identify aerosol microphysical properties. What this provides for is the means to atmospherically correct MODIS observations, and use that instrument's higher SNR and more spectral channels to determine the ocean state. A complete algorithm would probably utilize the VIS MISR channels as a verification that the retrieved ocean and atmosphere state is correct, and if not, be utilized in an iterative correction approach (Wang and Gordon, 1994, studied this prior to the launch of MISR). The scope of this paper is to verify that a single MISR NIR channel is sufficient to resolve the parameters traditionally used for AC. We find that it is, and then some.*

Regarding point (i), the aerosol microphysical properties are varied as described in table 1. Note that these aerosol models are used as such for typical AC retrievals, but in our case are free parameters related to aerosol 'relative humidity' (which modifies the refractive index and size of a given mode) and fine mode fraction (which governs the ratio of fine to coarse mode). This somewhat constrained parameterization is a requirement of the available information content, although it is less constrained than standard AC.

For point (iii) we found that the inclusion of sea foam in our simulations had negligible influence on information content, although proper handling would be required for AC. The Breon and Henriot reference regards the somewhat separate case of POLDER, which has access to the polarization state, more viewing angles, and different solar/view geometry, but we added that to the reference to Harmel and Chami in the introduction.

Minor comments:

The study is presented based on a few "test cases" corresponding to some AERONET- OC cases. First, those sites are mostly coastal with non-null NIR water-leaving radiance. Second, for such a theoretical study there is no need to restrict the analysis to very few and too specific conditions. For the sake of completeness, this test cases should be removed and replaced with a complete set of configurations, for instance sun angle from 0◦ to 90◦, aerosol optical thickness from 0 to 1, wind speed from 0 to 12 m/s (of course, actual values are at the discretion of the authors).

The reviewer has misunderstood our study, which we will attempt to resolve in the text. Our 'test cases' were chosen to simply identify specific geometries for which we know we already have a satellite – ground matchup. Beyond identifying real observed geometries (which span a range from high to low solar zenith angles) this offers the advantage that subsequent analysis with retrievals can be compared to this work. This study assessed retrieval capability for 1008 parameter combinations described in Table 2 for each of the seven geometries. Thus, Figures 3, 4, 5 and 7 represent specific geometry/parameter results, while figures 6, 8, 9 and 10 are represented as assessments in aggregate for many results.

Additionally, Figure 1 shows the relationship between observed solar zenith angle and relative azimuth angle. Both of these things strongly control the presence and location of the observed glint. We initially performed this study by stepping over a range of solar zenith and relative azimuth values but realized that it would be inclusive of geometries that are not, in fact, observed by MISR. We thus shifted gears and used real observation geometries instead.

While our approach was described in several points in the paper (such as section 2.6, implementation), we realize that reading section 2.3 could lead to misconceptions. So, we added a sentence to the end of that section to clarify.

Additionally, we have placed supplementary material with the results of each of these simulations as noted above.

Technical comments:
Through the manuscript: remove statement on future works, this gives the impression that everything is still to be done.

We removed these statements where it made sense to do so.

Title: specify the main purpose: atmospheric correction, aerosol retrieval. . . (see major comments)

We hope to have sufficiently address this as described above.

L.16: "virtually black"

We modified 'black' to 'strongly absorbing'

L.125: it would be very interesting to include more complex aerosol models than those obtained based on Mie assumptions (non-spherical, heterogeneous. . .)

Yes. But given our expectations based on previous information content assessments (ie Knobelspiesse, K., et al. Analysis of fine-mode aerosol retrieval capabilities by different passive remote sensing instrument designs, Opt. Express, 20(19), 21457-21484 , 2012.) we are unlikely to have the information content necessary to distinguish heterogeneous aerosols. Non spherical aerosols, on the other hand, are something we would like to address in an upcoming paper once we have incorporated that type of scattering into our radiative transfer model.

Table 1: specify the distribution type (in number, surface or volume) for the modal Parameters

We added a sentence in the Table 1 caption to note that the size distributions are log-normal, and referenced Ahmad et al 2010 from which the models are taken. It existed previously in section 2.1.

L.138. Provide the values of the increments used

We had this in the original manuscript but were unsure if it should be included. Now it is.

L.156: foam should be considered but if not you have to remove wind speed greater than a certain threshold (8, 10 or 12 m/s)

The largest assessed wind speed value was 7.49m/s, and as noted above inclusion of foam had no impact on our results. Foam would be considered in an actual retrieval.

Section 2.3: to be removed

We don't follow why this should be the case, unless it is referring to previous misunderstanding about the range of simulated values. We updated this section to hopefully clarify that issue.

L.199: "a less common extremely low θs(20◦)", why is it less common, actually we can have sun zenith angle = 0◦ for subtropical acquisitions.

Not with MISR, because it is on the Terra spacecraft in an inclined orbit with a 10:30am local equator crossing time. For example, in the 2020 summer solstice orbit, the minimum solar zenith angle was 14.71 degrees, maximum 80.91 degrees. The minimum values are larger at other times of the year, such that for the 2019 winter solstice the minimum value was 20.86 degrees, and 21.5 degrees at the 2020 spring equinox.

L.209: I would say: "the partial derivatives of the simulated signal in the vicinity of the retrieved parameters"

updated, thanks

L.211: in equation with δmj , mj shouldn't be bold.

Corrected, thanks

L.220: PDF not defined

This is, actually, defined in the introduction

L. 221: "measurement space is locally linear"? "locally continuous", instead?
Ok, that's better. Technically, we usually approximate the Jacobian with a forward difference calculation, which does mean locally linear, but we haven't gone into that detail in this manuscript.

L.253: it is not very clear to me, why eight dimensions?

The number of dimensions is n + n, where n is the number of parameters. We have a result for each 'node' in our lookup table (which has n dimensions), and each result is an n dimensional volume.

Table 4: σppis also a function of sun zenith angle

True, however, solar zenith angles where uncertainty has a meaningful contribution to total uncertainty are > 70.

References

Breon, F. M., Henriot, N. (2006). Spaceborne observations of ocean glint reflectance and modeling of wave slope distributions. Journal Of Geophysical Research-Oceans, 111, C06005.

Harmel, T., Chami, M. (2013). Estimation of the sun glint radiance field from optical satellite imagery over open ocean: multidirectional approach and polarization aspects. Journal of Geophysical Research, 118(1), 1–15. https://doi.org/10.1029/2012JC008221

Munk, W. (2009). An Inconvenient Sea Truth: Spread, Steepness, and Skew- ness of Surface Slopes. Annual Review of Marine Science, 1(1), 377–415. https://doi.org/10.1146/annurev.marine.010908.163940

Thank you for these. We added Munk and Breon & Henriot, Harmel and Chami was already listed.

**Referee: Hongqing Liu**

This manuscript has an interesting topic of assessing the information content of the MISR multi-angle measurements aiming for a potential simultaneous retrieval of aerosol and wind speed. By using AFRT to generate a lookup table and applying GENRA for ICA, several sensitivity tests are designed to reveal the optimal choices of the retrieval strategy. I think this is a good paper, and I don't have any disagreements regarding the analysis and conclusions; but I do have some questions about the technique, especially how to implement the GENRA approach and perform the sensitivity test.

My general understanding (not sure it is right) of this work is that, instead of using real MISR data, the simulated MISR measurements for given sets of parameters (r,f,$\tau$ ,w) at the geometries of the test cases (interpolated from the LUT) would be used as "inputs" to derive the posterior PDF of the parameters (r',f',$\tau$',w'), where the prime indicates the "retrieval space" which is much finer than that of the "input" (as in the LUT), so "the result will have n+n dimensions" (line 253-254).

My first question is for equation 2: what is the "likelihood function"? It is said "$Pd(y)$ is the stochastic measurement distribution (d for data), and $Pl(y)$ is the same for the likelihood function" (line 243). Is the "$Pl$" has the same function form as "$Pd$" (equation 12), but using different y'? From Vukicevic's paper (2010), her equation 1 has the stochastic measurement distribution ($Pd$) and model distribution ($Pt$) terms, and the "likelihood" is the "combining of the model and measurement pdfs" (section 2.3), which seems to correspond to the "$Pd(y)$" only, since "the data ($Pd$) is a PDF created from a single node in the LUT and expectations of measurement and model uncertainty" (line 248-249), as defined in the equation 12, the uncertainty variance ($\sigma$) includes both the measurement and model contributions. In this paper, the "likelihood function" is mentioned at five places, but it is hard to me to get a clear understanding of this term.

Thank you for these thoughts, and we agree with you that this is a subtle topic, often difficult to transition from the comparatively simple theory to practical application (i.e. code). By way of explanation, consider the likelihood function ($p_l$) in the absence of model uncertainty. In that case, the function (a PDF) is a delta function with no width. If we assume that the prior ($p_r$) is uninformative, then the posterior ($p_o$) for a given m is the probability that the data correspond to the simulation from the likelihood function.

I attempted to draw a cartoon to illustrate this process. Again, making the assumption that the prior is uninformative (has no effect), and treating the likelihood as a delta function (no model uncertainty), then the top row represents four different summations from equation (2), at different parameter values (m). In each, the 'data' are the same, as illustrated by the light blue Gaussian. The radiative transfer model output for a given parameter value (m1, m2, m3 or m4) is the likelihood delta function indicated by different colors for each parameter value. $p_d$ and $p_l$ are multiplied, then the summation is made over all values of y (data). The result is the $p_o$ for each parameter value as indicated by the plot at the bottom. As you can see, we start to get an idea of what the shape of the a posteriori PDF should look like, and parameter value m2 is the most likely value so far given the observation.

The shape of the 'data' PDF $p_d$ comes from the observed value (or in our case, a simulation node in our LUT) and the expected uncertainty (equation 12). In practice, we use a sigma (defining the width of $p_d$) that is a (squared) summation of all sources of uncertainty, both measurement and model. It makes the code operation more efficient to keep $p_l$ as a delta function, and is mathematically identical. The sparsity of $p_o$ also shows why we must heavily interpolate the calculations that generate $p_l$, so that $p_o$ is not overly coarse.

Of course, our illustration is for a one dimensional cartoon, in practice this occurs over the four dimensional parameter space. Also, one view angle observation is assessed at a time, and $p_o$ becomes the prior for subsequent assessments.

Hopefully this illustration clarifies the application of GENRA.

[Figure]

My second question is about the last two sensitivity tests (sections 3.4 and 3.5). It appears to me (again, might be wrong) that the simulated MISR measurements (from LUT) are calculated with the plane-parallel and scalar wind speed assumptions. It is a little hard to understand how to test the sensitivity to the conditions whose signals are not in the inputs. For the test of the plane parallel assumption (section 3.4), if the simulated Df and Da camera observations are from the plane parallel calculations, then it would be expected that retrieval with them (9-

camera) would have a better result than without (7-camera), since the (extra) information contained in these two cameras are consistent with the retrieval assumptions (both assume plane parallel). For the sensitivity test of the glint, discarding the uncertainty associated with vector wind would only lead to the improvement since the inversion would be more consistent with the input (both assume the scalar wind without penalty). If I didn't misinterpret the result, figure 10 seems indicate that the test without the scalar uncertainty term would have more information content than the test with the uncertainty since the SIC difference is positive.

Taking the above illustration as an example, we performed a GENRA analysis with larger model uncertainties, and compared that to the standard GENRA results. In both cases we estimated a model uncertainty, either from the literature (as was the case with the plane parallel assessment) or by testing the differences between a more complex model and a simpler one (as was the case for the scalar vs vector wind speed). In both analysis we used the same model calculations, but with larger model uncertainty. For the plane parallel case, we also did not incorporate the Da and Df camera observations.

So, you are correct that, without the vector based uncertainty in figure 10, the information content is higher. This indicates that the use of the scalar instead of the vector model does have consequences. However, in most cases (top panel of Figure 10) the difference is quite small, and the cases where it matters are limited to low solar zenith angles (bottom panel). Considering that an additional parameter adds computation expense and retrieval ambiguity, we decided that the difference was not significant enough to warrant a change. This is further illustrated in Figure 11.

Hopefully this answers your questions.

Other than these two questions, I think the paper is well written and the analysis is thorough and clear. So I would recommend this paper be accepted after some clarification about the above questions are made.

Thank you.

One minor issue, for the equations 4-7, should the summation be make over the parameters with prime (i.e., r', f' etc.)?

Actually, no. Consider that we are attempting to reduce an 8 dimensional volume to five dimensions – one for each of the simulation nodes, and the other representing the marginal PDF for a given parameter at that node. Thus summations over three dimensions reduces the volume dimensionality from 8 to 5.

**Anonymous Referee #1**
This study presents a theoretical information content analysis to evaluate the capability of multi-angular MISR 865nm measurements to simultaneously determine aerosol and ocean surface sun glint variables, including AOD, fine-mode fraction, relative humidity, and surface

wind speed. Such analysis was accomplished by a radiative transfer simulation of seven sets of MISR observation geometries and the Bayesian-based information content (and error) analysis. Authors also performed several comparative information analysis to evaluate the sensitivity of retrieval errors to solar & view geometries, plane-parallel atmosphere assumption in radiative transfer, and wind direction.

This is an important study, as indicated by the authors, which is to lay the groundwork for the design of an algorithm for a simultaneous retrieval of aerosol and ocean surface properties from MISR. The paper is well written and well organized. I have the following comments that author may consider:

Thank you

1. It seems to me that correlation between errors of individual observations and correllation between the prior error of each state elements are not considered in the Bayesian information analysis. However, these error covariances may exist between MISR's multi-angular observations, as well as between some retrieved variables (such as AOD and fine-mode fraction, AOD and RH). It is thus necessary to explicitly declare the assumption of error correlation in this work and discuss its potential impacts to the results.

This is true, although we expect the impact to be minimal for MISR. To further clarify, we added the following statement to the end of section 2.4:

*Finally, we should note that a disadvantage of our implementation of GENRA is that it does not account for uncertainty correlation in the data. However, we expect MISR observations for each camera to be largely uncorrelated (Kahn et al. (2005b)). The resulting posterior PDF expresses covariance between parameters, but cannot account for model uncertainties that are correlated in parameter space.*

2. I agree with the reviewer #1 that more analysis should be performed to extend the currently selected geometrical conditions. It will be of interest to see the results for MISR observations in solar principal plane and perpendicular plane (which are currently missing in the selected geometries, as in Figure 1).

Please refer to the comments above regarding this topic. We would like to reiterate that MISR does not observe in the solar principal plane, and observes in the perpendicular plane at low solar zenith angles (roughly 20 degrees), where the relative azimuth angle is relatively unimportant. Case F is about 20 degrees from the perpendicular plane.

For an illustration of all MISR observed geometries in an orbit, please see Figure 2, panel (b) of Knobelspiesse and Nag, 2018 (https://doi.org/10.5194/amt-11-3935-2018). Analogous to Figure 1 in this paper, we can see that observation geometries close to the solar principal plane only occur at extremely high solar zenith angles (corresponding to high latitudes), while the

perpendicular plane is observed at low zenith angles (about 20 degrees, corresponding to low latitudes).

3. In the section of results, readers need to compare posterior PDFs for different conditions but tend to get lost to flip back-and-forth between Figures 3 to 7. I would recommend to rearrange the panels in Figure 3-7 by combining same panels for different conditions. Doing so will help the comparative interpretation.

We agree that visualization of high dimensional spaces is difficult, and we much balance utility and brevity. To that end, we don't want to repeat figures reconstructed in multiple formats in the same manuscript. However, we have taken the marginal PDF's from the three comparable cases (same parameter state, different geometry) and rearranged them in a figure here. Since this reviewer response is also available to all readers, we hope this is an adequate solution.

[Figure]

4. The symbol "r" is used to represent both relative humidity and particle radius. Authors may want to use different symbols for them to avoid confusion.

Good point. We updated particle radius in equation 1 to use 'R' rather than r.

---

## Referee Report (RR1)

Thanks for the author's response, I think this paper can be published as is with one extra request: please remove my name from the acknowledgements since I didn't have any constructive contributions to improve this paper. The author can just thank three reviewers together.

Below is my reply to the author's response. Please don't take this as further comments asking for author's extra work, this is just an exchange of opinions.

First, thanks for the cartoon, which is really good to illustrate the process of GENRA. For my first question, it appears to me that "$P_d$" (Equation 12) alone might be enough for the likelihood with both the measurement and model uncertainties included, and I am still curious about how to calculate (or what is) the value of the "delta function $p_l$" if it is really needed as described in the implement step 2-(b)-ii-D (section 2.6).

For the second question, my concern is that if the same tests (with larger model uncertainties or the standard GENRA) are applied to the real MISR measurements (not the model simulation with the plane parallel and scalar wind assumptions), is there any possibility that we might get different results since the inputs are very different -- one (the measurements) with the non-plane-parallel and vector wind effects, and another (model simulation) without?

For the Equations 4-7, I am still not sure why the variables to be marginalized out (under the summation operators on the RHS of the equation) still appear on the LHS of the equation? It makes more sense to me that the primed variables (r',f'..) should be marginalized, i.e., $p_m^r(r', r, f, \tau, w) = \sum_{f'} \sum_{\tau'} \sum_{w'} p_o(m)$

I have to admit that my concerns might be totally wrong, and I need to think more about them to get a better understand of this technique. Maybe I can get the answers later when I have a chance to study the code after the paper is published. Please forgive my ignorance if my concerns do not make sense.